# Motor context dominates output from purkinje cell functional regions during reflexive visuomotor behaviours

**Laura D Knogler, Andreas M Kist, Ruben Portugues\***

Max Planck Institute of Neurobiology, Sensorimotor Control Research Group, Martinsried, Germany

**Abstract** The cerebellum integrates sensory stimuli and motor actions to enable smooth coordination and motor learning. Here we harness the innate behavioral repertoire of the larval zebrafish to characterize the spatiotemporal dynamics of feature coding across the entire Purkinje cell population during visual stimuli and the reflexive behaviors that they elicit. Population imaging reveals three spatially-clustered regions of Purkinje cell activity along the rostrocaudal axis. Complementary single-cell electrophysiological recordings assign these Purkinje cells to one of three functional phenotypes that encode a specific visual, and not motor, signal via complex spikes. In contrast, simple spike output of most Purkinje cells is strongly driven by motor-related tail and eye signals. Interactions between complex and simple spikes show heterogeneous modulation patterns across different Purkinje cells, which become temporally restricted during swimming episodes. Our findings reveal how sensorimotor information is encoded by individual Purkinje cells and organized into behavioral modules across the entire cerebellum.
DOI: https://doi.org/10.7554/eLife.42138.001

## Introduction

Decades of influential anatomical (*Eccles et al., 1967*; *Palay and Chan-Palay, 1974*), theoretical (*Marr, 1969*; *Albus, 1971*; *Ito, 1972*) and experimental work (see *Ito, 2006* for review) have led to our current knowledge highlighting the cerebellum as a major brain region for the control of motor behaviors. This ability to coordinate motor control and learning relies critically on the integration of sensory and motor-related signals in Purkinje cells, as these neurons constitute the main computational units and output of the cerebellum. In order to understand the detailed operations of the cerebellum, it is therefore of fundamental importance to characterize the physiology of cerebellar neurons, especially the Purkinje cells, during sensorimotor behaviors.

Purkinje cells receive two excitatory input streams, via parallel fibers from granule cells and a single climbing fiber from the inferior olive, that differentially modulate their spike output. Across vertebrate species, climbing fibers from inferior olivary neurons drive complex spikes in Purkinje cells at a spontaneous rate of ~0.5–2 Hz whereas parallel fiber inputs modulate intrinsic simple spike activity at much higher rates (from tens of Hz in larval zebrafish up to hundreds of Hz in mammals; *Hsieh et al., 2014*; *Eccles et al., 1967*; *Raman and Bean, 1997*). Simple spike output can furthermore be biased to burst or pause by the arrival of a complex spike (*Mathews et al., 2012*; *Badura et al., 2013*; *Sengupta and Thirumalai, 2015*), though the precise nature of this relationship varies across Purkinje cells (*Zhou et al., 2014*; *Zhou et al., 2015*; *Xiao et al., 2014*). In addition, inhibitory interneurons may also exert considerable control over simple spike rates (*Dizon and Khodakhah, 2011*; *ten Brinke et al., 2015*; *Jelitai et al., 2016*). Characterizing the type of information carried by these different input streams at the population level and disentangling their relative contributions to Purkinje cell output has been challenging due to the large number of Purkinje cells in

**\*For correspondence:**
rportugues@neuro.mpg.de

**Competing interests:** The authors declare that no competing interests exist.

the mammalian cerebellum (>300,000 in the rat cerebellum) receiving convergent input from 100,000 to 200,000 parallel fibers (*Harvey and Napper, 1991*).

Due to this complicated physiology, the anatomy of climbing fiber projections onto Purkinje cells has primarily been used to characterize the organization of the mammalian cerebellum. Four transverse zones along the rostrocaudal axis have been described (*Ozol et al., 1999*) that can be further subdivided into longitudinal zones and microzones defined by additional anatomical, physiological and molecular features (see *Apps and Hawkes, 2009* for review). This organization is thought to produce functional modules that each participate in the control of a certain set of behaviors (*Cerminara and Apps, 2011*). However, since these regions have been largely defined in terms of anatomical rather than physiological properties, the behavioral relevance of cerebellar modules is not well understood. Purkinje cells are at the center of cerebellar circuits, integrating climbing fiber and parallel fiber input. A detailed description of the flow of sensory and motor information within both individual and groups of Purkinje cells is important to understand the functional significance of these proposed behavioral modules.

A fundamental first step is therefore a population-level investigation of Purkinje cell activity during a simple set of sensorimotor behaviors with single-cell resolution of simple and complex spikes. In this study, we took advantage of the larval zebrafish to study how sensorimotor variables are encoded in Purkinje cell output during reflexive, visually-driven motor behaviors. The larval zebrafish cerebellum is anatomically organized in a typical vertebrate tri-layered configuration, with a population of fewer than 500 Purkinje cells, each receiving inputs from many parallel fibers and likely just one climbing fiber (*Bae et al., 2009*; *Hashimoto and Hibi, 2012*; *Hsieh et al., 2014*; *Hamling et al., 2015*). Several studies have demonstrated a functional role for the cerebellum in the larval zebrafish relating to motor coordination, adaptation, and learning (*Aizenberg and Schuman, 2011*; *Ahrens et al., 2012*; *Matsui et al., 2014*; *Portugues et al., 2014*; *Harmon et al., 2017*). The behavioral repertoire of the larval zebrafish includes robust but variable swimming and eye movements to drifting gratings and rotating stimuli (the optomotor and optokinetic response, respectively). These visual stimuli are particularly useful because they elicit graded, episodic swim bouts and eye movements that vary across trials, allowing us to disambiguate clearly between sensory and motor responses. We are furthermore able to extract many different features from both the visual stimuli and motor behaviors (i.e. onset, direction, velocity) to pinpoint how Purkinje cell activity correlates with particular features of visual stimuli at a fine temporal scale.

Using this approach, in this study we investigated three main questions: (1) how motor and sensory information is encoded in individual Purkinje cells from different input pathways, (2) how the temporal dynamics of these different information streams are encoded in Purkinje cell output, and (3) how responses are spatially organized across the entire cerebellum. Calcium imaging across the whole cerebellum to the same set of visual stimuli in tandem with tail-free and eye-free behavior revealed considerable spatial segregation in Purkinje responses. We supplemented calcium imaging data with direct electrophysiological recordings in order to examine complex and simple spikes directly under conditions of fictive or eye-free behavior. In agreement with our imaging data, we uncovered a consistent and striking organization of the Purkinje cell population into three functional regions along the rostrocaudal axis that encode visual information with respect to either directional motion onset, rotational motion velocity, or changes in luminance. The fine temporal resolution of our electrophysiological recordings together with our ability to disentangle different sensorimotor variables revealed that these regions receive similar motor-related parallel fiber input but are strongly differentiated by sensory complex spike responses that encode distinct visual features with unique temporal dynamics. We relate these findings to other work in the field to propose an overarching organization of the larval zebrafish cerebellum into cerebellar modules underlying innate and flexible visually-driven behaviors.

## Results

### Activity in the cerebellum is arranged into functionally-defined and anatomically-clustered symmetrical regions of Purkinje cells

Anatomical, physiological, and genetic studies of the mammalian cerebellum across species show that the cerebellar cortex is organized into spatially-restricted regions of Purkinje cells, where a

given region has a specific set of inputs and outputs and is thought to control the coordination and adaptation of a different set of sensorimotor behaviors (*Apps and Hawkes, 2009*; *Witter and De Zeeuw, 2015*). In order to describe the organization of Purkinje cell responses across the entire cerebellum with high spatial resolution, we performed two-photon calcium imaging across the complete population of Purkinje cells while presenting a variety of visual stimuli that drive variable, reflexive sensorimotor behaviors (*Easter and Nicola, 1996*) to awake, head-embedded larval zebrafish whose eyes and tail were freed and could move (*Figure 1a,b*; see *Video 1* for an animation of visual stimuli as presented to the fish during two-photon imaging experiments).

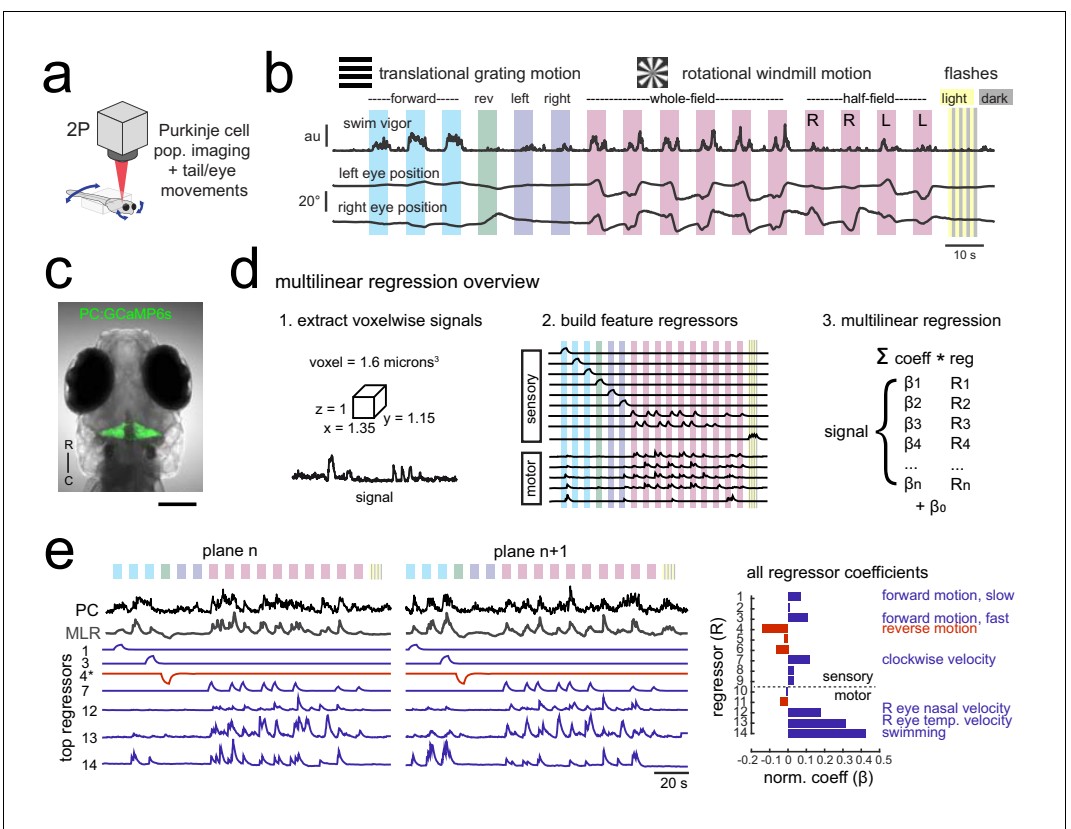

**Figure 1.** Using population imaging and multilinear regression to describe feature responses across the Purkinje population during visuomotor behaviors. (a) Cartoon of the embedded zebrafish preparation under the two-photon microscope with freely-moving eyes and tail. (b) Overview of the visual stimuli presented to the awake, behaving zebrafish during volumetric two-photon calcium imaging. See Materials and methods for further details. The mean swimming activity and eye position for a representative fish across an entire experiment is shown (N = 100 trials). (c) Composite bright field image of a seven dpf zebrafish larva from a dorsal view showing Purkinje cells expressing GCaMP6s driven by a ca8 enhancer element. Scale bar = 100 microns. (d) Overview of the multilinear regression analysis. See Materials and methods for additional details and see *Figure 1—figure supplement 1* for full list of regressors. (e) Left panels, example calcium signal from a Purkinje cell across two planes (black trace) can be well recapitulated through multilinear regression (MLR, grey trace; $R^2$ = 0.77). The regressors with the seven largest coefficients ($\beta$) are shown below scaled in height and colored by their $\beta$ value (blue = positive, red = negative). The asterisk for regressor four refers to a negative value of $\beta$ which results in an inverted regressor. Right, a bar graph quantifying the normalized $\beta$ values for all regressors for this cell with the regressors shown at left labelled. See also *Figure 1—figure supplements 1* and *2*.
DOI: https://doi.org/10.7554/eLife.42138.002

The following figure supplements are available for figure 1:

**Figure supplement 1.** Functional imaging anatomy and full regressor list.
DOI: https://doi.org/10.7554/eLife.42138.003

**Figure supplement 2.** Calcium signals report complex spikes reliably but can also report simple spike bursts.
DOI: https://doi.org/10.7554/eLife.42138.007

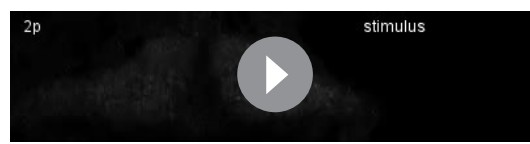

**Video 1.** Z-projection map of GCaMP6s responses (max dF/F) in Purkinje cells to visual stimuli. Related to *Figure 2*.
DOI: https://doi.org/10.7554/eLife.42138.004

We observed frequent eye and tail movements that varied across visual stimuli and across trials (*Figure 1b*). Whole-field gratings moving in the four cardinal directions elicited reflexive but variable optomotor swimming responses. Swimming episodes (bouts) were evoked in a probabilistic manner that was modulated by the direction and speed of the visual motion (i.e. no swim response to gratings in the reverse direction). A windmill pattern centered on the larva's head rotating with a sinusoidal velocity elicited a reflexive optokinetic response of the eyes that also showed some behavioral variability across trials. Moderate intensity whole-field flashes were included to provide stimuli that evoke no acute behavioral response (*Figure 1b*) but that could nonetheless contribute to ethological behaviors over longer timescales, for example relating to circadian rhythms (*Burgess and Granato, 2007*). The visual stimuli were presented in open loop (i.e. with no updating of the visual stimuli in response to behavior) in order to clearly dissociate the sensory stimuli and any behavioral response. It should be noted that visually-driven motor behaviors are robust on average but episodic and variable across trials, allowing us to clearly disambiguate sensory and motor contributions to neuronal activity when we examine the correlations between Purkinje cell activity and eye or tail motor activity on a trial by trial basis.

We used two-photon calcium imaging to image neural activity in 7 days-post fertilization (dpf) zebrafish larvae expressing GCaMP6s in all Purkinje cells (*Figure 1c*, *Figure 1—figure supplement 1a*). This strategy allowed us to measure the entire Purkinje cell population in response to this set of stimuli with high spatial resolution while tracking eye and tail movements. Neural responses to these

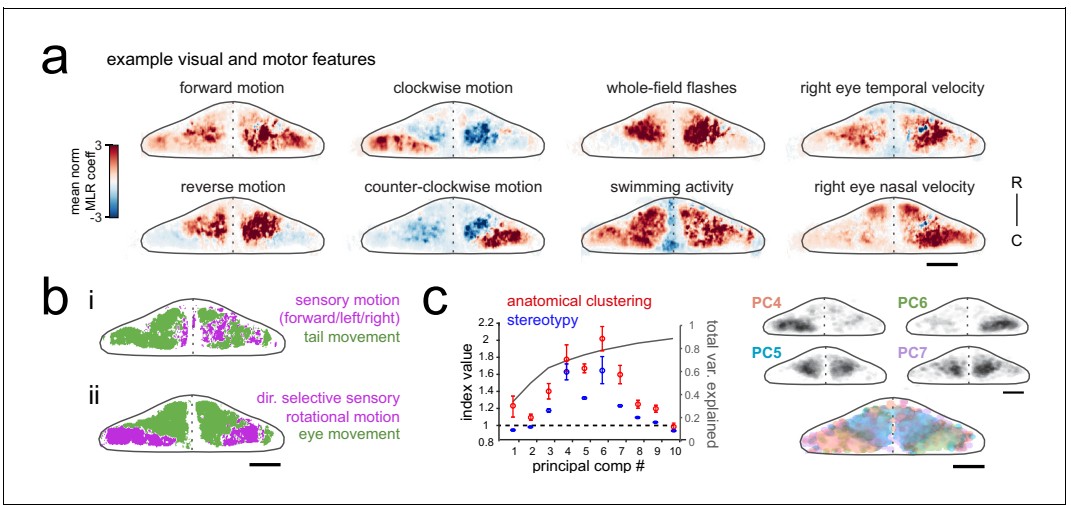

**Figure 2.** Purkinje cell activity is functionally clustered across the cerebellum. (**a**) Heatmaps of the z-projected mean voxelwise correlation coefficients from multilinear regression (MLR) with example sensory and motor regressors for a representative fish (see Materials and methods). Scale bar = 50 microns. (**b**) Voxels from the example fish in a) are colored according to whether the best regressor for correlated sensory stimuli and motor events (including i) swimming and ii) eye movement) are sensory (magenta), motor (green), or equal/uncorrelated (white). (**c**) Left, quantification of principal component analysis, clustering, and stereotypy of Purkinje cell responses. Left axis, index values across the first ten principal components with respect to the anatomical clustering of principal components within a fish (red line) and the stereotypy of these clusters across fish (blue line). Dotted black line shows an index value of 1 (equivalent to chance). Right axis, total variance explained across principal components. Right panel, mean spatial mapping of the four principal components with the highest index values for anatomical clustering and stereotypy as individual maps (above) and composite (below). Colors are arbitrarily chosen.
DOI: https://doi.org/10.7554/eLife.42138.006

stimuli showed considerable temporal and spatial structure across the cerebellum, as visualized by the z-projection map of average calcium responses (max dF/F) in the entire Purkinje cell population across the trial (*Video 1*) as well as in the activity from single imaging planes (*Video 2*). We estimated the number of Purkinje cells in the larval zebrafish to be $433 \pm 19$ (mean $\pm$ std, N = 3) by identifying spherical nuclei in confocal stacks of a transgenic line that expresses nuclear-localized GCaMP6s using 3D template matching (*Figure 1—figure supplement 1*; see Materials and methods). This number is higher than the previously reported range of 180–360 Purkinje cells at seven dpf (N = 6; *Hamling et al., 2015*).

In order to quantify how different features of the visual stimuli and the tail and eye behaviors contribute to Purkinje cell activity, we performed multilinear regression on voxelwise calcium signals obtained across the Purkinje cell population (*Figure 1d*, *Figure 1—figure supplement 1*; see Materials and methods for detailed description). Multilinear regression is advantageous for two reasons in particular. First, it allows the identification of multiple visual and/or motor features that may contribute to a single calcium signal. Second, we can distinguish between regressors that may be moderately correlated in our experiments, such as forward moving gratings and the variable swim bouts that these stimuli elicit. Zebrafish swim in episodic bursts of swimming that last just hundreds of milliseconds, separated by rest periods lasting seconds, whereas the visual stimuli driving these swim bouts were presented for many seconds. As a result, motor regressors for eye or tail movements look very different from visual sensory regressors (*Figure 1d,e*, *Figure 1—figure supplement 1*) and their respective contributions to calcium signals can be determined.

Our analysis showed that Purkinje cell activity is functionally segregated across the cerebellum with respect to different visual and motor features (*Figure 2a*, *Video 3*). Responses to whole-field flashes were enriched in a bilaterally symmetric central region of the cerebellar cortex, whereas responses to clockwise and counterclockwise rotational motion had an asymmetric localization within the left and right hemisphere of the caudolateral cerebellum, respectively. Purkinje cell responses to motor activity, including eye and tail motion, were generally broad and showed strong, uniform correlations across most of the cerebellar cortex.

Next, to disambiguate between visual and motor responses we explicitly visualized the sensory/ motor preference across the Purkinje cell layer for two visuomotor behaviors: swimming driven by forward-moving gratings, and left/right eye movements driven by rotational windmill motion. *Figure 2b* shows a z-projection of the cerebellum for each of these visuomotor behaviors, with areas colored magenta or green based on whether the relevant visual or motor feature was significantly better in explaining the activity in that region (see Materials and methods). As *Figure 2bi* shows, the activity of Purkinje cells distributed across a broad region of the cerebellum correlated highly with tail movement during swimming and accounted for the modulation of calcium activity to a much greater extent that sensory grating motion. In contrast, *Figure 2bii* shows that a large dense bilateral area of the caudolateral cerebellum had activity that was more strongly related to sensory rotational motion while the remaining area of the rostral and medial cerebellum showed a stronger correlation with eye movements. These results indicate that locomotor activity of the tail and eyes is broadly encoded in Purkinje cell activity across the cerebellum whereas sensory responses to visual features are more anatomically clustered.

Finally, in order to identify groups of Purkinje cells whose activity was similarly modulated during the experiment, regardless of which feature

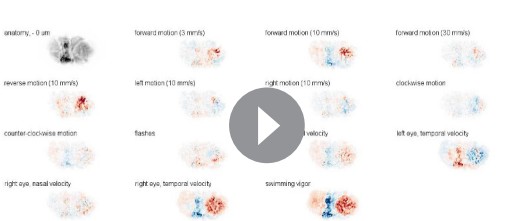

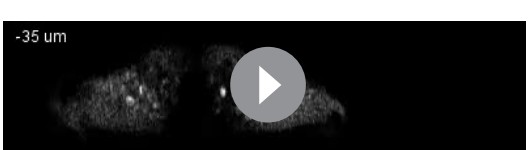

**Video 2.** Single plane at −35 microns depth from the dorsal surface showing GCaMP6s responses (max dF/F) in Purkinje cells to visual stimuli. Related to *Figure 2*.
DOI: https://doi.org/10.7554/eLife.42138.005

**Video 3.** Upper left, anatomical stack of Purkinje cell anatomy (upper left) showing the depth in microns of the plane from the dorsal surface. Other panels, the corresponding plane from the stack of regressor coefficient weights (labelled for regressor type) for all Purkinje cells as quantified with multilinear regression (see Materials and methods). Related to *Figure 2*.
DOI: https://doi.org/10.7554/eLife.42138.008

drove the response, we performed principal component analysis on the coefficient weights for all cells across all fish (N = 10; see Materials and methods). This analysis again revealed considerable spatial structure and stereotypy in Purkinje cell responses, with most functional clusters being also both anatomically clustered within fish and similarly located across fish (*Figure 2c*, *Figure 1—figure supplement 1d*). Four functional clusters emerged that were particularly spatially-clustered and tiled the cerebellum across the rostrocaudal axis of each hemisphere (*Figure 2c*). Together, these results suggest a clear spatial organization of Purkinje cells into functional regions along the rostrocaudal axis of the zebrafish cerebellum.

## Calcium signals in Purkinje cells report complex spikes with high fidelity with lesser contributions from simple spikes

Since Purkinje cells receive excitatory inputs from both climbing fibers and parallel fibers that drive different types of spiking, it is critical to understand exactly what the calcium signals described above represent in terms of the underlying spike identity and structure. Climbing fiber inputs driving complex spikes have been shown to reliably produce large dendritic calcium signals in mammalian Purkinje cells with little to no signal in the soma (*Lev-Ram et al., 1992*). In contrast, parallel fiber inputs may contribute to small, local calcium signals at dendritic spines or branchlets (see *Kitamura and Kano, 2013* for review) while changes in sodium-dependent simple spike rates may be read out from somatic calcium signals (*Ramirez and Stell, 2016*). We performed in vivo cell-attached electrophysiological recordings of spontaneous activity from single Purkinje cells expressing GCaMP6s in order to show how the signals obtained during calcium imaging relate to complex and simple spike output in larval zebrafish Purkinje cells (*Figure 1—figure supplement 2*).

As expected, we found that every complex spike elicited a peak in the calcium signal of a Purkinje cell's dendrites (*Figure 1—figure supplement 2a,b*). However, we also found that isolated bursts of simple spikes correlated with widespread increases in the dendritic calcium signal. Aligning the calcium signal to the onset of simple spike bursts and single complex spike events showed consistent simple spike-triggered calcium transients that were of smaller amplitude but similar duration to complex spike-triggered transients (*Figure 1—figure supplement 2b,d*). We used multilinear regression methods to determine the relative contribution from the activity of complex and simple spikes to the calcium signals we measured in different Purkinje cells across fish. We find that although the majority of the signal is driven by the occurrence of a complex spike, simple spikes also contribute to a varying degree across cells and can account for up to half of the calcium signal (mean percentage contribution from complex spikes = 78.4 ± 6.8%, N = 8 cells from eight fish; *Figure 1—figure supplement 2e,f*).

These findings reveal that both complex spikes and simple spike bursts can contribute to the dendritic fluorescence signals obtained by calcium imaging in larval zebrafish Purkinje cells. The observation above that many visual and motor features can contribute to the calcium signal from a single Purkinje cell (*Figure 1e*) is therefore unsurprising if this signal represents not only complex spikes but also simple spike responses modulated by the convergent input from many parallel fibers. We furthermore observed that somatic signals and dendritic signals were highly correlated with each other (mean correlation = 0.87 ± 0.2, N = 5 cells from three fish; *Figure 1—figure supplement 2g, h*), suggesting that the contribution from these different input streams may not be as spatially segregated in these Purkinje cells as shown in other systems and therefore cannot be isolated by subcellular imaging. In summary, calcium signals across Purkinje cells report both complex spikes and high frequency simple spiking and care must therefore be taken when interpreting the underlying activity patterns of Purkinje cells measured with functional imaging.

## Electrophysiological recordings from Purkinje cells reveal distinct complex spike responses that can be grouped into three primary visual response phenotypes

In order to overcome the mixed contribution of complex spikes and simple spike bursts to calcium signals and to record Purkinje cell spiking activity in greater detail, we turned to single-cell electrophysiology. We performed cell-attached Purkinje cell electrophysiological recordings at different locations across the cerebellum in the awake, paralyzed larval zebrafish while presenting visual stimuli as for the functional imaging experiments described above (N = 61 cells from 61 fish). Complex

spikes and simple spikes can be clearly distinguished in these recordings with automated thresholding by amplitude (*Figure 3a*) and converted to a spike rate (*Figure 3b*; see Materials and methods). Simultaneous fictive recordings of locomotor activity were obtained from a ventral root extracellular electrode (*Figure 3a*) as previously described (*Masino and Fetcho, 2005*) and used to extract information about fictive swim bouts (see Materiials and methods). The high temporal resolution of electrophysiological recordings further enhances our ability to separate feature components. For example, we find that swimming activity is only moderately correlated with forward visual motion on a trial by trial basis (mean correlation = 0.31 ± 0.2, *Figure 3c*, *Figure 3—figure supplement 2*).

In an approach similar to that used to analyze the functional imaging results presented above, we built regressors to capture the most salient features of the visual and motor stimuli (see *Figure 3d* for examples and *Figure 3—figure supplement 1* for the full regressor list). The high temporal resolution of electrophysiology allows us to resolve transient changes in simple spike firing rate as well as single complex spikes, therefore we added regressors for the visual and motor regressors that would capture spiking responses to a more specific set of visual stimulus and behavioral features such as visual motion onset, duration, velocity, swim onset, and graded swim strength. The window chosen for stimulus onset covered 500 milliseconds from actual stimulus onset (e.g. motion onset of forward gratings) in order to account for the inherent synaptic delays for visual information to arrive in the cerebellar input layer, on the order of 100–200 milliseconds (*Knogler et al., 2017*). Preliminary assessments of spike rates with visual and motor feature regressors further confirmed that these regressors appropriately captured the temporal dynamics of Purkinje cell spiking (*Figure 3d*, *Figure 3—figure supplement 2*). We employed a variant of multilinear regression with elastic net optimization that includes regularization terms to help sparsify the number of features that are used to reconstruct the signal, as well as variable selection and parameter optimization to overcome the minor degree of correlation between some regressors (*Figure 3e* and *Figure 3—figure supplement 1*; see Materials and methods).

Since the complex spikes and simple spikes of Purkinje cells are modulated by climbing fiber and parallel fiber input streams, respectively, we independently assessed these responses across the population of cells (*Figure 3e*). We will first address the complex spike responses, as these provided a useful classification of Purkinje cell groups within the population in line with the functional and spatial organization seen during functional imaging.

We observed that complex spikes were generally evoked by a narrow subset of stimuli. Only a few visual or motor features provided a significant contribution to each cell's complex spike rate (mean number of nonzero coefficients = 6.0 ± 0.4 out of 22, N = 61), and in many cases a single feature was very dominant (*Figure 3e*). Mixed complex spike responses to multiple stimuli are possible due to mixed selectivity in neurons of the inferior olive (*Ohmae and Medina, 2015*; *Ju et al., 2018*) or residual multiple climbing fiber input (*Crepel et al., 1976*). We found little evidence however that individual Purkinje cells encode multiple types of visual stimuli or both visual and motor features in their complex spike responses. The current results do not rule out the likely possibility that information from other sensory modalities than vision are also encoded in the complex spikes of these cells.

A survey of the best regressor category for each cell from this dataset revealed that Purkinje cell complex spike responses were strongly enriched for visual information (*Figure 3e*), specifically the onset of direction-specific translational motion (N = 31/61) and direction-specific rotational velocity (N = 14/61). The remaining Purkinje cells were categorized as having complex spikes that best responded to changes in whole-field luminance, to fictive motor activity, or to the duration of translational motion. Notably, sensory responses across visual features are far better represented than motor responses in the complex spike responses of Purkinje cells (*Figure 3e*). This was not due to a paucity of motor activity, as bouts of swimming behavior were consistently elicited across trials. Only 8/61 cells had the biggest contribution to complex spike rates from motor activity, and across the remaining cells the average contribution from motor regressors was less than 5% (3.7 ± 1%, N = 53). Of the eight cells whose best regressor was motor-related, there were nonetheless significant responses to visual features present as determined by non-zero sensory coefficient weights accounting for 10–40% of the complex spike activity (mean contribution = 20 ± 5%). As a result, we made the surprising observation that all but one of the Purkinje cells that we recorded from across the entire cerebellum could be unambiguously assigned to one of three visual complex spike 'phenotypes' corresponding to a response to directionally-selective translational motion onset, directionally-selective rotational velocity, or changes in luminance.

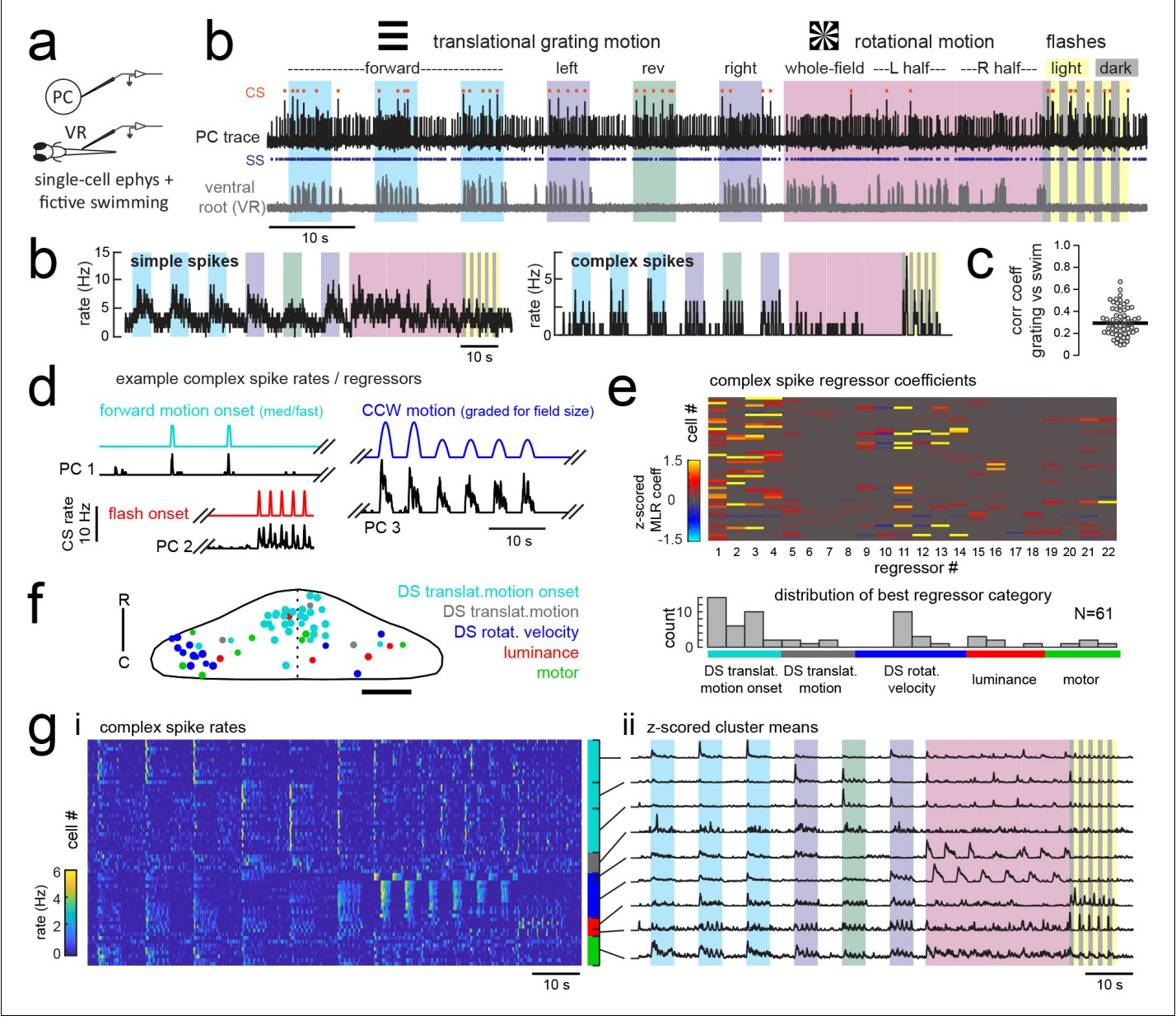

**Figure 3.** Electrophysiological recordings from Purkinje cells reveal distinct complex spike responses that can be grouped into four primary response types corresponding to sensory or motor features. (a) Cartoon of the embedded, paralyzed zebrafish preparation used for simultaneous Purkinje cell (PC) electrophysiology with fictive swimming patterns extracted from the ventral root (VR). (b) Example single trial from a cell-attached Purkinje cell (PC) recording (upper trace, black) with simultaneous ventral root recording (lower trace, gray, shown as a moving standard deviation). Complex spikes in the PC are indicated by orange dots above the trace and simple spikes are indicated by blue dots below the trace. Stimuli are color-coded as before (see *Figure 1* and Materials and methods for more details). (b) Left, the mean simple spike (SS) and complex spike (CS) rate for the cell shown in (a) across five trials. Right, the correlation coefficients of forward, left and rightward grating motion with the trial by trial fictive swim activity for all fish. (c) Plot of the correlation coefficient for each fish between the regressor for concatenated swimming activity during moving forward, left, and right gratings across all trials and the summed sensory regressor for forward, left, and right grating motion. The mean is indicated by the black bar. (d) Example mean complex spike rate extracts from three different Purkinje cells showing the temporal similarity of firing dynamics with visual feature regressors. (e) Above, heatmap of coefficient weights for the complex spike firing rates of 61 cells from z-scored least-squares multilinear regression (MLR) with a full set of 24 stimulus- and motor-related variables (see Materials and methods). Below, histogram showing the distribution of cells' highest regressor weight. (f) Location of these cells across all fish mapped onto a reference cerebellum (dorsal view). The color indicates the highest MLR coefficient weight for that cell while the size indicates the degree to which that coefficient contributes to the overall firing rate respective to the others, where the biggest circles = 100%. Scale bar = 50 microns. (g) Left, heatmap of complex spike rates for all 61 cells clustered according to the category of their

*Figure 3 continued on next page*

*Figure 3 continued*

highest MLR coefficient weight (e.g. luminance, rotational motion, swimming). Colored bars at right indicate complex spike category as indicated in previous panels. Right, the mean z-scored complex spike rate from each cluster. See also *Figure 3—figure supplements 1* and *2*.

DOI: https://doi.org/10.7554/eLife.42138.009

The following figure supplements are available for figure 3:

**Figure supplement 1.** Sensory and motor regressors used for multilinear least-squares regression with electrophysiological recordings Top left, cartoon of recording setup.

DOI: https://doi.org/10.7554/eLife.42138.010

**Figure supplement 2.** Visually-evoked swimming responses to forward gratings are episodic, vary across trials, and are clearly resolvable from visual responses.

DOI: https://doi.org/10.7554/eLife.42138.011

We hypothesized that these three different visual complex spike phenotypes could underlie the spatial clustering of Purkinje cell population activity that we observed with functional imaging (*Figure 2c*). Mapping the coordinates of all Purkinje cells onto a reference cerebellum revealed a spatial organization of complex spike sensory response phenotypes similar to our functional imaging data (*Figure 3f*). In particular, we observed a rostromedial cluster of cells responsive to the onset of directional motion in the visual stimulus and a caudolateral cluster of cells responsive to rotational stimulus velocity. Luminance responses were more scattered but generally occupied the central zone between these regions.

Together with our functional imaging data, these results suggest that zebrafish Purkinje cells contribute to the formation of three distinct spatial regions across each cerebellar hemisphere through visual complex spike profiles encoding either directionally-selective translational motion onset, directionally-selective rotational velocity, or changes in luminance. These regions bear a striking resemblance to the anatomically clustered activity patterns identified by principal component analysis in our imaging data (*Figure 2c*), suggesting that the visual complex spike response phenotype is an important parameter that can be used to understand the spatial and functional organization of Purkinje cells across the cerebellum.

## Purkinje cells in different regions receive feature-specific climbing fiber input and project to different downstream regions

From the three major visual complex spike phenotypes we observed across the Purkinje cell population, we observed that further subdivisions could be made based on the specific type of response to a given visual stimulus. For example, direction-selective motion onset-responsive Purkinje cells differ in their directional tuning, and luminance-responsive cells can prefer either increases or decreases in luminance, or bidirectional changes (*Figure 3d,g*). Therefore, we next performed further detailed analyses of Purkinje cell complex spike activity in combination with additional anatomical experiments in order to quantify precisely how visual features such as directionality are encoded by different Purkinje cells with the same visual phenotype and to also identify the projection patterns of Purkinje cells across phenotypes.

The largest group of Purkinje cells showed a phenotype for strong, direction-selective responses to the onset of translational motion (N = 33/61 cells). These responses typically spanned two of the four cardinal directions tested, producing on average just one complex spike at the onset of motion in the preferred directions (1.2 ± 0.6 spikes/stimulus; *Figure 4a*). The occurrence of a complex spike was not dependent on the behavioral response since visually-evoked complex spikes occurred with equal probability whether there was a swimming response or not (*Figure 3—figure supplement 2*). In the clearest example, reverse visual motion evokes no swimming but is equally well-represented by a complex spike response at motion onset as the directions that do drive swimming (*Figures 3g* and *4a*, *Figure 3—figure supplement 2a*).

The direction selectivity index (see Materials and methods) of these cells ranged from 0.2 to 0.9 (*Figure 4—figure supplement 1a*), and cells typically responded to two of the four cardinal directions tested (*Figure 4a*). No cells were found that responded significantly to motion onset in opposing directions. Although the Purkinje cell somata displaying this complex spike phenotype were closely clustered in the most rostromedial part of the cerebellum (*Figure 3f*), the lateralization of Purkinje cells was biased such that cells in the left cerebellar hemisphere preferred either forward

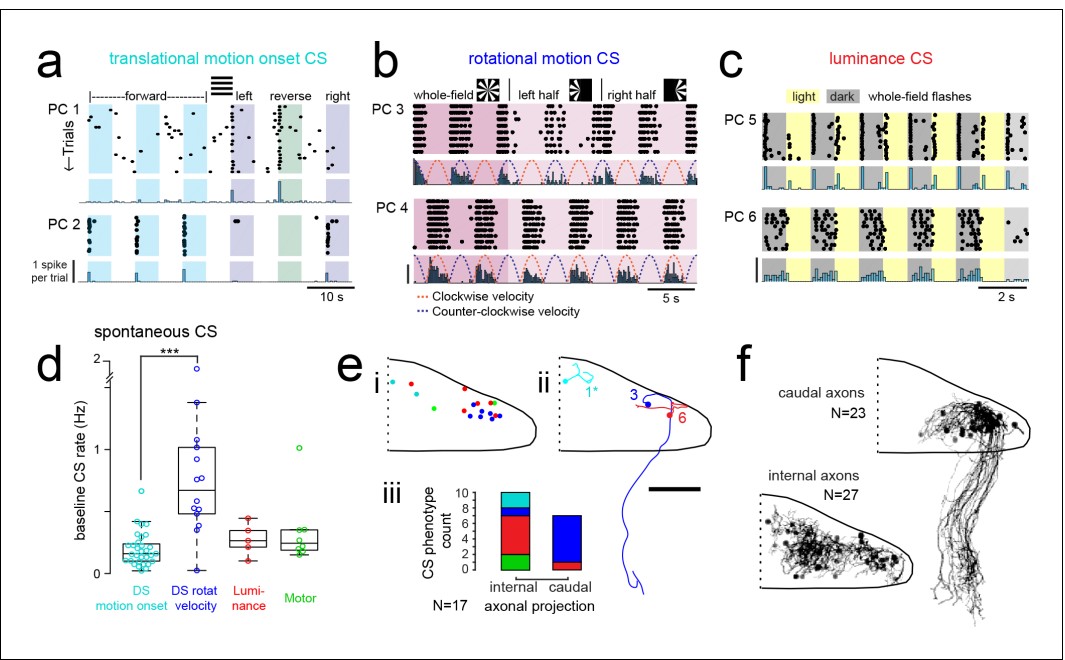

**Figure 4.** Purkinje cells in different regions show complex spike responses that encode different visual features and one group sends outputs to a different downstream region. (**a**) Raster plot (upper left panels) and histogram (lower left panels, 500 ms bins) of complex spikes occurring across trials during translational whole-field motion of black and white bars in all four cardinal directions for two example Purkinje cells (PC). Numbers assigned to PCs for this and panels b-c are arbitrary. (**b**) Raster plot (upper left panels) and histogram (lower left panels, 100 ms bins) of complex spikes occurring across trials during whole- and half-field bidirectional rotational motion of a black and white windmill for an example cell. The dashed lines over the histogram show the velocity of the stimulus in each direction across the trial. (**c**) Raster plot (upper left panels) and histogram (lower left panels, 100 ms bins) of complex spikes occurring across trials during whole-field light/dark flashes for two example cells, (i) and ii). (**d**) A box plot of complex spike firing rates during blank trials (no visual stimuli) for cells grouped by their sensory or motor complex spike category (see *Figure 2*). N = 31, 14, 5, 8. Asterisks indicate significance (one-way ANOVA with Bonferroni post hoc correction, p<0.001). j (i) The location of cells colored by complex spike phenotype are plotted onto a flattened dorsal view of the cerebellum with all coordinates flipped to the right half of the cerebellum. e (ii) Three example maximum projection images of traced axonal morphology from stochastically-labelled, Fyn-mClover3-expressing Purkinje cells for which electrophysiological recordings were also obtained. Labels for each cell refer to the electrophysiological traces in panels a-c. The asterisk for cell a) indicates that these coordinates were flipped to the right half of the cerebellum. Scale bar = 50 microns. e (iii) Categorical grouping of complex spike phenotypes for internal versus caudal axonal projections. N = 17 cells from 17 fish. (**f**) Morphed Purkinje cell axonal morphologies from single-cell labelling across fish (N = 50 cells) can be grouped into two populations based on axonal projection (as for e iii). N = 27 cells with internal axons, N = 23 cells with caudal axons. See also *Figure 4—figure supplement 1*.

DOI: https://doi.org/10.7554/eLife.42138.012

The following figure supplements are available for figure 4:

**Figure supplement 1.** Complex spike responses encode specific aspects of visual features.

DOI: https://doi.org/10.7554/eLife.42138.013

**Figure supplement 2.** Purkinje cell dendrites show a mostly planar morphology.

DOI: https://doi.org/10.7554/eLife.42138.014

**Figure supplement 3.** Motor-related complex spikes are rare.

DOI: https://doi.org/10.7554/eLife.42138.015

motion to the right (0 to 90°, N = 7) or reverse motion to the left (−90 to −180°, N = 5; *Figure 4—figure supplement 1a*). Conversely, Purkinje cells in the right cerebellar hemisphere preferred either forward motion to the left (0 to −90°, N = 5) or reverse motion to the right (90 to 180°, N = 5; *Figure 4—figure supplement 1a*). The reliable, phasic nature of these complex spike responses suggests that these Purkinje cells encode acute, directional changes in the visual field.

The second group of Purkinje cells, located in the caudolateral cerebellum, showed a phenotype for large, directionally-selective increase in complex spikes during either clockwise or counter-clockwise rotational motion that unlike the previous group persisted throughout the duration of movement (*Figure 4b*; N = 12/61 cells). During rotational motion in the preferred direction, complex spike firing rates in these cells were two to five times higher than baseline (mean rate increase = 340 ± 40%, N = 12). In contrast, complex spike rates during motion in the non-preferred direction fell to nearly zero, well below the baseline rate (mean rate non-preferred direction = 0.32 ± 0.1 Hz versus 0.87 ± 0.1 Hz at baseline; *Figure 4—figure supplement 1b*). Consistent with our functional imaging data (*Figure 2*), these complex spike responses to rotational motion were highly lateralized such that all Purkinje cells (10/10) that preferred clockwise rotational motion were located in the caudolateral region of the left cerebellar hemisphere while the only two Purkinje cells that preferred counterclockwise motion were located in the mirror symmetric region of the right cerebellar hemisphere (*Figure 4—figure supplement 1b*). Additional experiments in the semi-paralyzed animal (see 'Motor-related complex spikes are rare for tail and eye movements', below) confirmed this laterality (N = 11; *Figure 4—figure supplement 3h*).

These Purkinje cells also showed an increase in complex spiking for the duration of translational motion in a preferred lateral direction, determined to be rightwards motion for clockwise motion-preferring cells and vice versa (mean rate increase above baseline = 280 ± 20%; *Figure 4—figure supplement 1b*), suggesting that these cells respond to motion over a large area situated in the front half of the visual field. Finally, we observed an apparent homeostatic regulation firing in these cells where spontaneous complex spike rates were strongly depressed for several seconds following the robust complex spike responses elicited by rotational motion (normalized mean rate for one second following rotational stimuli = 37 ± 16% of spontaneous firing rates). Thus a high complex spike rate for the preferred direction of rotational motion may come at the expense of stochastic complex spikes. A homeostatic regulation of complex spike rates has also recently been observed in the mammalian cerebellum (*Ju et al., 2018*), though the underlying mechanism is not known.

The third prominent group of Purkinje cells had complex spike responses correlated with changes in whole-field luminance that were surprisingly heterogeneous in feature encoding compared to the notably stereotyped responses seen for the previous two groups (N = 25/61 cells; *Figure 4c* and *Figure 4—figure supplement 1c–e*; see Materials and methods). Purkinje cells with this luminance phenotype had complex spike responses that encoded either luminance increases (9/25) or decreases (11/25) or both (5/25; *Figure 4c* and *Figure 4—figure supplement 1d*) and the location of cells with different luminance response types was mixed across the central region of the cerebellum (*Figure 4—figure supplement 1e*). The latency from the onset of the preferred luminance transition to the first complex spike occurred for each cell with very little jitter, but the latency itself varied across cells (*Figure 4—figure supplement 1d*). Most whole-field luminance responses were transient such that cells fired just one complex spike for the preferred luminance change (mean = 0.80 ± 0.02 spikes). We did however observe, in two cells, different complex firing rates as a function of the ambient luminance presented that did not adapt over tens of minutes and therefore appear to encode ambient luminance through their complex spike rate (*Figure 4cii*, one-way ANOVA with Bonferroni post hoc correction, p<0.01; *Figure 4—figure supplement 1g,h*). Luminance-responsive Purkinje cells furthermore showed differing patterns of complex spike responses to local luminance changes during the translational motion of gratings (*Figure 4—figure supplement 1f*), suggesting that these cells have a wide range of receptive field sizes over which they integrate luminance.

In additional to having qualitative and quantitative differences in visual feature encoding, the three different types of Purkinje cell visual phenotypes described thus far also had notable differences in spontaneous complex spike rates (*Figure 4d*). Purkinje cells responding to rotational motion velocity had a significantly higher baseline firing rate than those with directionally selective motion onset responses (0.77 ± 0.1 Hz and 0.20 ± 0.02 Hz, respectively, p<0.001, Bonferroni post hoc correction). Purkinje cells responding most strongly to luminance or motor activity had intermediate baseline complex spike rates (0.28 ± 0.1 Hz and 0.34 ± 0.1 Hz, respectively).

Mapping the coordinates of Purkinje cell somata belonging to these three visual complex spike phenotypes supports a regional division of the cerebellum along the rostrocaudal axis where each of the three regions within the cerebellar hemisphere receives inputs from the same or similar inferior olive neurons carrying visual information (*Figure 4ei*). To examine the corresponding outputs of

Purkinje cells from these groups, we performed cell-attached electrophysiological recordings in combination with morphological reconstructions via stochastic single-cell labelling of Purkinje cells to visualize the axonal projections (N = 17 cells from 17 fish; *Figure 4eii*). Unlike the mammalian cerebellum, where all Purkinje cell axons project outside of the cerebellar cortex, zebrafish Purkinje cells can be divided into two anatomical populations - one with internally-projecting axons that contact eurydendroid cells (the equivalent of mammalian cerebellar nuclei neurons) and the other whose somata are more lateral and who have externally-projecting caudal axons that contact neurons in the vestibular nuclei (*Bae et al., 2009*; *Matsui et al., 2014*).

Strikingly, 6/7 Purkinje cells with caudally projecting axons exhibited a clear complex spike phenotype for directional rotational motion, whereas only 1/10 cells with an internal axon had this same phenotype (*Figure 4e*). We further reconstructed and aligned 50 singly-labelled Purkinje cell morphologies across fish to a reference brain. Although the somata of Purkinje cells with caudal (N = 23/50) and internal (N = 27/50) axons partially overlap (*Figure 4f*), the segregation of rotational motion responses with caudal axon anatomies in this dataset further support our definition of this functional region of Purkinje cells. We also found that Purkinje cell dendrites generally had a classic albeit simplified morphology with mostly planar dendrites (*Figure 4—figure supplement 2*) oriented orthogonally to the axis of parallel fiber extension across the cerebellum (*Knogler et al., 2017*), as seen in mammalian cerebellum (*Eccles et al., 1967*). Together, these results define three functional groups of Purkinje cells residing in different regions across the cerebellum. These groups operate with different complex spike frequencies and encode distinct visual features related to visuomotor behaviors, and one group also sends the majority of its projections to a different downstream area than the others.

## Motor-related complex spikes are rare for tail and eye movements

As discussed above, motor regressors did not significantly contribute to complex spike activity in the majority of Purkinje cells (N = 49/61) despite an abundance of visually-evoked fictive behavior and the use of multiple motor regressors to capture different motor features. We nonetheless used this small group of Purkinje cells with motor-related complex spike responses to examine motor feature encoding (*Figure 4—figure supplement 3*).

We analyzed complex spike responses from Purkinje cells during spontaneously-evoked swimming in blank trials as well as in trials where visual stimuli elicited swimming and confirmed that some complex spikes were indeed correlated with swimming activity even in the absence of visual stimuli (*Figure 4—figure supplement 3a*). Swim-related responses were however unreliable such that a complex spike occurred on fewer than half of swim bouts on average for these cells (mean probability = 0.38 ± 0.07 for stimuli trials, N = 9 cells; mean probability for blank trials = 0.42 ± 0.07, N = 5 cells; p<0.05, Wilcoxon signed rank test). Aligning the subset of swim bouts that were positive for motor-related complex spikes showed that the latency from bout onset to the occurrence of a complex spike in blank trials varied considerably for an individual cell, in contrast to the fixed latencies for most visual-driven complex spikes (*Figure 4—figure supplement 3b*, compare with *Figure 4* and *Figure 3—figure supplement 2*). This is consistent with observations that complex spikes do not show phase-locking with stereotyped locomotor movements (*Apps, 1999*). Some Purkinje cells showed a decrease in complex spikes during swim bouts with a subsequent increase following bout offset (mean probability of a complex spike during bout <0.02, N = 3 cells; *Figure 4—figure supplement 3c*) however this was rarer than those with motor-related increases (*Figure 4—figure supplement 3d*). Unlike the spatial mapping seen for Purkinje cells with visual complex spike responses, Purkinje cells with motor-related complex signals were distributed across the cerebellum with no apparent clustering (*Figure 4—figure supplement 3e*).

We observed that both translational and rotational visual motion induced frequent bouts of fictive swimming in fish (*Figure 3b*); however the complex spike responses during these visual stimuli in most Purkinje cells did not correlate well on a trial-by-trial basis with swim bouts (*Figure 3b*, *Figure 3—figure supplement 2*) and were thus classified from multilinear regression analysis as sensory (visual), as described above. Rotational windmill stimuli are however known to evoke stereotyped eye movements known as the optokinetic reflex (*Easter and Nicola, 1996*), therefore complex spike responses to rotational visual motion could relate to the activation of eye rather than tail muscles. Studies of the cerebellar control of eye movements have shown evidence that climbing fibers provide eye motor error signals, which could account for the prominent complex spike signals observed

in Purkinje cells in the caudolateral cerebellum during rotational windmill motion. In order to examine the potential contribution of eye movements to complex spikes in this group of Purkinje cells, we performed cell-attached recordings from Purkinje cells in the caudolateral cerebellum in the semiparalyzed zebrafish, where the eyes were free to move and were tracked with a high-speed camera (see Materials and methods). The independent movement of each eye was then used to build a set of twelve regressors corresponding to eye position and velocity in different directions (*Figure 4—figure supplement 3f*).

Least-squares multilinear regression was used to analyze the complex spike activity of all cells with the existing set of sensory regressors for visual features and the twelve new eye motor regressors. We once again observed a clear bias for a visual complex spike phenotypes across Purkinje cells (N = 11/13), with only two cells whose best regressor related to eye movement (*Figure 4—figure supplement 3g,h*). A further analysis of the complex spike phenotypes of these latter two cells showed that in one case, the eye movement was exceptionally well-correlated with one visual feature (directional rotational velocity) that it was hard to disambiguate the true sensory vs motor nature of the complex spike response (*Figure 4—figure supplement 3i*, cell 2). For the other cell (*Figure 4—figure supplement 3i*, cell 3), a strong luminance (sensory) complex spike phenotype was identified through additional autocorrelation analyses (p<0.001 from Ljung-Box Q-test; r = 0.67 for correlation with the luminance regressor) in addition to the moderate correlation with eye movement (r = 0.32 for correlation with eye motor regressor). Nonetheless, the majority of Purkinje cells in this region could be clearly assigned to a visual complex spike phenotype since these cells showed very stereotyped complex spike responses to directional rotational stimuli that did not correlate well with the variable eye movements observed across trials (*Figure 4—figure supplement 3j,k*). We conclude from these data that the complex spike responses during rotational visual motion are predominantly sensory rather than motor. It is furthermore important to note that these responses are equally prominent in electrophysiological recordings in the paralyzed fish, where the eyes cannot move, as in the electrophysiological and functional imaging experiments where the eyes are free and track the stimulus (compare *Figure 2*, *Figure 3f–g*, and *Figure 4—figure supplement 3*).

## Simple spike responses across the Purkinje cell population are highly modulated by motor efference copies during fictive swimming

Having observed that Purkinje cells can be clustered into functional regions defined by their visual complex spike responses and anatomical features, we next wanted to understand how simple spike responses were organized across the cerebellum. Multilinear regression showed that many visual and motor features contributed to simple spike responses in individual Purkinje cells, such that response phenotypes were broader than those observed for complex spike activity (*Figure 5a* and *Figure 3e*), as expected in a circuit where many parallel fibers converge on a single Purkinje cell (*De Zeeuw et al., 2011*). Although simple spike rates were modulated to some extent by many of the visual stimuli presented, motor activity significantly modulated simple spike activity in nearly all Purkinje cells across the cerebellum (N = 60/61) and was in fact the main contributor to modulating simple spike activity in the majority of cells (N = 44/61; *Figure 5a*).

Different motor regressors accounted for various motor features including swim onset, offset, duration, and the continuous quantitative readout of swim strength, termed vigor (calculated from the standard deviation of the ventral root signal). Simple spike firing rates for these cells had consistently larger contributions to their activity from swim vigor than from bout duration or any other motor regressor, suggesting that fictive swimming activity is encoded in a graded manner by simple spike output. Mean simple spike firing rates across the population were on average twice as high during a bout as during the rest of the trial (mean rate during a bout = 14.5 ± 1.5 Hz vs 7.6 ± 0.8 Hz at rest; p<0.001, Wilcoxon signed rank test). Trial-averaged simple spike responses across the population appeared as a continuum rather than as clusters (*Figure 5b*), suggesting that the organization of parallel fiber inputs does not follow the same regional specificity as climbing fiber inputs across the cerebellum. Our analyses furthermore revealed that translational and rotational motion of visual stimuli, regardless of direction, was the most prominent sensory feature encoded by simple spike activity (*Figure 5a*). These findings suggest that Purkinje cells are integrating inputs from motion responsive granule cells with different directional tuning (*Knogler et al., 2017*).

In order to rule out potential sensory contributions to simple spike rates during visually-evoked behaviors, we analyzed simple spike activity during additional blank trials where no visual stimuli was

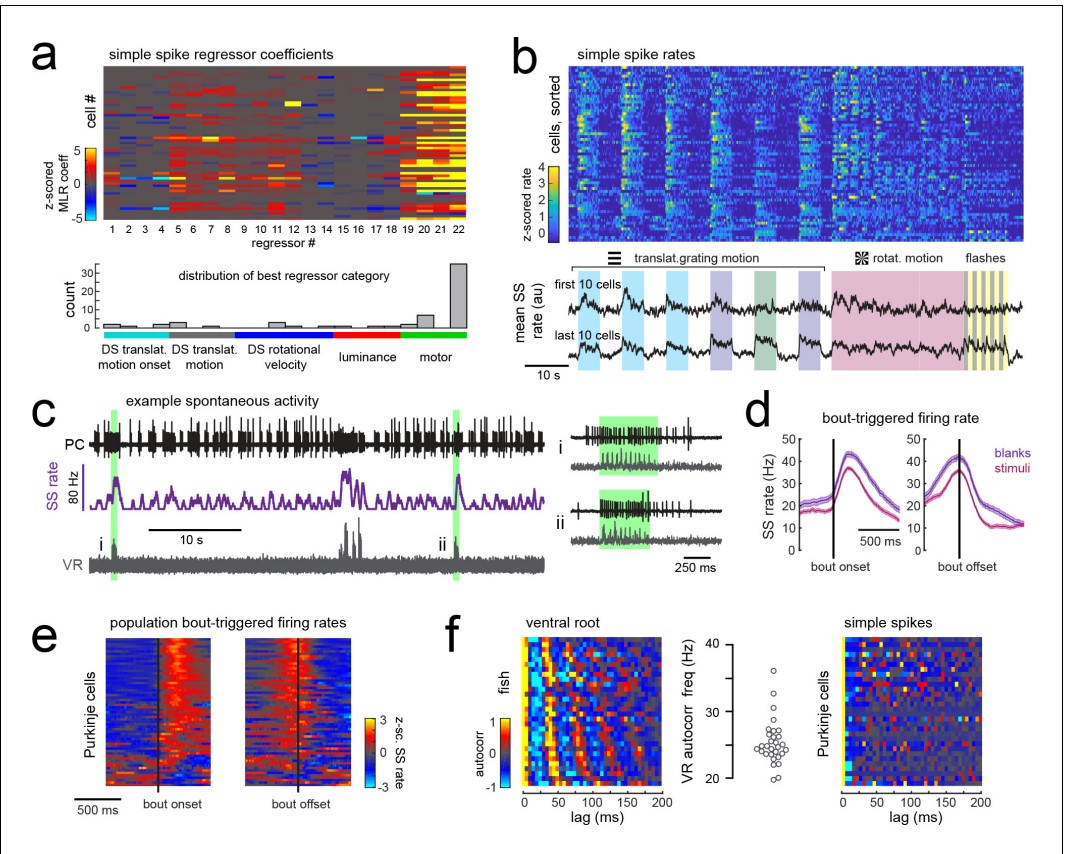

**Figure 5.** Simple spike rates in most Purkinje cells are increased during fictive swimming. (**a**) Above, heatmap of coefficient weights for the simple spike firing rates of 61 cells from least-squares regression with a full set of 24 stimulus- and motor-related variables (see Materials and methods for more details). Below, histogram showing the distribution of cells' highest regressor weight and the associated sensory/motor categories. (**b**) Upper panel, heatmap of z-scored simple spike rates for all 61 cells sorted by decreasing motor coefficient weight. Lower panel, the mean simple spike rate for the ten cells with the highest (upper trace) and lowest (lower trace) motor coefficient weights. (**c**) Left panel, example cell-attached Purkinje cell recording (PC, upper trace, black) from a blank trial (no stimuli) with simultaneous ventral root recording (VR, lower trace, gray, shown as a moving standard deviation). The simple spike rate is also shown (SSrate, middle trace, purple). Right, the bouts highlighted in green on an expanded timescale show the close timing of fictive bout onset and simple spike activity. (**d**) The bout on- and off-triggered mean simple spike firing rates for the cell in c) during blank recordings (purple) and stimulus trials (pink). (**e**) Z-scored heatmap of bout on- and off-triggered mean simple spike firing rates across all Purkinje cells sorted by mean firing rate in the 300 ms following bout onset. (**f**) Mean autocorrelation heatmap for simple spikes (SS, upper panel) and for ventral root recordings (VR, lower panel) for all Purkinje cells that showed spontaneous swimming bouts during blank trials (N = 30 cells from 30 fish), sorted by time to first peak in the VR autocorrelation. Right, the first significant peak in the VR autocorrelation for each recording is plotted to give the mean fictive swim frequency for each fish.

DOI: https://doi.org/10.7554/eLife.42138.016

presented (*Figure 5c*). Purkinje cells exhibit considerable spontaneous simple spike firing in the absence of any sensory stimuli or motor activity (*Hsieh et al., 2014*; *Sengupta and Thirumalai, 2015*); however fictive swim bouts consistently increased simple spike firing well above baseline levels (*Figure 5c–e*) and to an ever greater extent for spontaneous bouts in the absence of visual stimuli (p<0.01, Wilcoxon signed rank test). Aligning the mean bout-triggered simple spike rates for all Purkinje cells at bout onset and offset confirmed that the majority of cells have consistent motor-related increases in simple spike activity that begin at bout onset and return to baseline following bout offset (*Figure 5e*) although a small number of Purkinje cells instead show bout-triggered decreases in simple spike firing rates (*Figure 5e*), as observed elsewhere (*Scalise et al., 2016*). As expected for

rhythmic locomotor output, the ventral root signal was highly autocorrelated for all fish with a mean autocorrelation frequency across fish of 26.7 ± 0.7 Hz (N = 30 fish; *Figure 5f*), consistent with the slow swim bout frequency reported for restrained zebrafish larvae (*Severi et al., 2014*). The autocorrelation analyses of simple spike firing for each Purkinje cell during a spontaneous fictive bout revealed however no significant autocorrelations for simple spikes at any frequency. Unlike the modulation of simple spike firing rate seen during step phase in locomoting rats (*Sauerbrei et al., 2015*), simple spikes in zebrafish Purkinje cells do not appear to be modulated in phase with rhythmic swimming activity but are nonetheless graded by swim strength. This suggests that individual Purkinje cell firing does not encode the activation of individual muscles involved in rhythmic swimming.

## Motor activity is broadly represented in granule cell signals

The timing and reliability of swim-related simple spike activity is consistent with motor efference signals from spinal locomotor circuits during fictive swimming that arrive along mossy fibers to the granule cell layer and subsequently to Purkinje cells. A disynaptic pathway from spinal premotor interneurons to the granule cells via the lateral reticular nucleus was recently found that would convey information about ongoing network activity in the spinal cord (*Pivetta et al., 2014*). There is growing evidence across species that granule cells are strongly driven by ongoing locomotor activity (*Ozden et al., 2012*; *Powell et al., 2015*; *Jelitai et al., 2016*; *Giovannucci et al., 2017*; *Knogler et al., 2017*). Furthermore, extensive electrophysiological recordings from the granule cells of the cerebellum-like circuit of the electric organ in the electric fish revealed that an overwhelming majority (>90%) of granule cells receive depolarizing motor efference signals during electric organ discharge, although this translated into spiking in only ~20% of granule cells (*Kennedy et al., 2014*).

In order to characterize motor-related granule cell activity and its potential contribution to motor-related excitation in Purkinje cells across the cerebellum, we imaged responses in the granule cell population to the same set of visual stimuli while tracking tail and eye movement (*Figure 6a*). Multi-linear regression was once again used to disambiguate responses to sensory stimuli and motor activity. Across fish (N = 7), we observed that granule cell activity was strong and widespread during swimming activity, both in the somatic layer and across the parallel fiber layer (*Figure 6a*). Granule cell activity relating to eye movements was weaker but also widespread (*Figure 6a*). These findings suggest that a large number of granule cells receive mossy fiber inputs relaying motor efference copies that drive them to fire, and they in turn drive broad motor-related activation of simple spikes in Purkinje cells (*Figure 2a,b*). These findings show more widespread motor-related representations in comparison to previous population-wide analyses of granule cell activity (*Knogler et al., 2017*) due to the abundance of behavior elicited by the current set of visual stimuli.

In order to understand the temporal patterning of swim-related motor signals in the granule cells layer, we performed additional electrophysiological recordings from randomly targeted granule cells across the cerebellum while simultaneously recording ventral root activity to identify fictive swim episodes. These recordings revealed several granule cells with negligible firing rates in the absence of motor activity but that showed large, significant increases in their spike rates during fictive bouts (N = 6/8 cells; mean firing rate at rest = 1.3 ± 0.3 Hz vs 25.7 ± 7.6 Hz during a bout; p<0.005, Wilcoxon signed rank test; *Figure 6b–d*; *Figure 6—figure supplement 1*). These granule cells had graded responses correlated with swim strength and could reach high instantaneous firing frequencies of up to 150 Hz during a fictive bout, similar to the burst firing seen in mammalian granule cells during locomotion (*Powell et al., 2015*) or whisker stimulation (*van Beugen et al., 2013*). Half of these motor-excited granule cells (N = 3/6) also showed significant autocorrelations in their spiking activity during fictive swimming (p<0.001, Ljung-Box Q-test; *Figure 6e*). The frequency of the spike autocorrelations for these cells was comparable to the fictive swim frequency obtained from the ventral root (mean difference in frequency = 1.3 ± 0.6 Hz, N = 3), suggesting that the periodicity of granule cell spiking is related to the swimming activity (*Figure 6e*). The phase of the granule cell spiking with respect to the ipsilateral ventral root activity varied however across cells, arriving either in phase, with a lag, or in antiphase (*Figure 6f*).

Together, these results suggest that motor efference copies are relayed along mossy fibers to many granule cells to drive burst firing during swimming bouts, whether fictive or real. In turn, parallel fibers deliver graded swim-related excitation to nearly all cerebellar Purkinje cells. We are confident that these are true efference signals and not motor-related sensory input from proprioception or the lateral line since the fish is paralyzed and the muscles are not moving during these

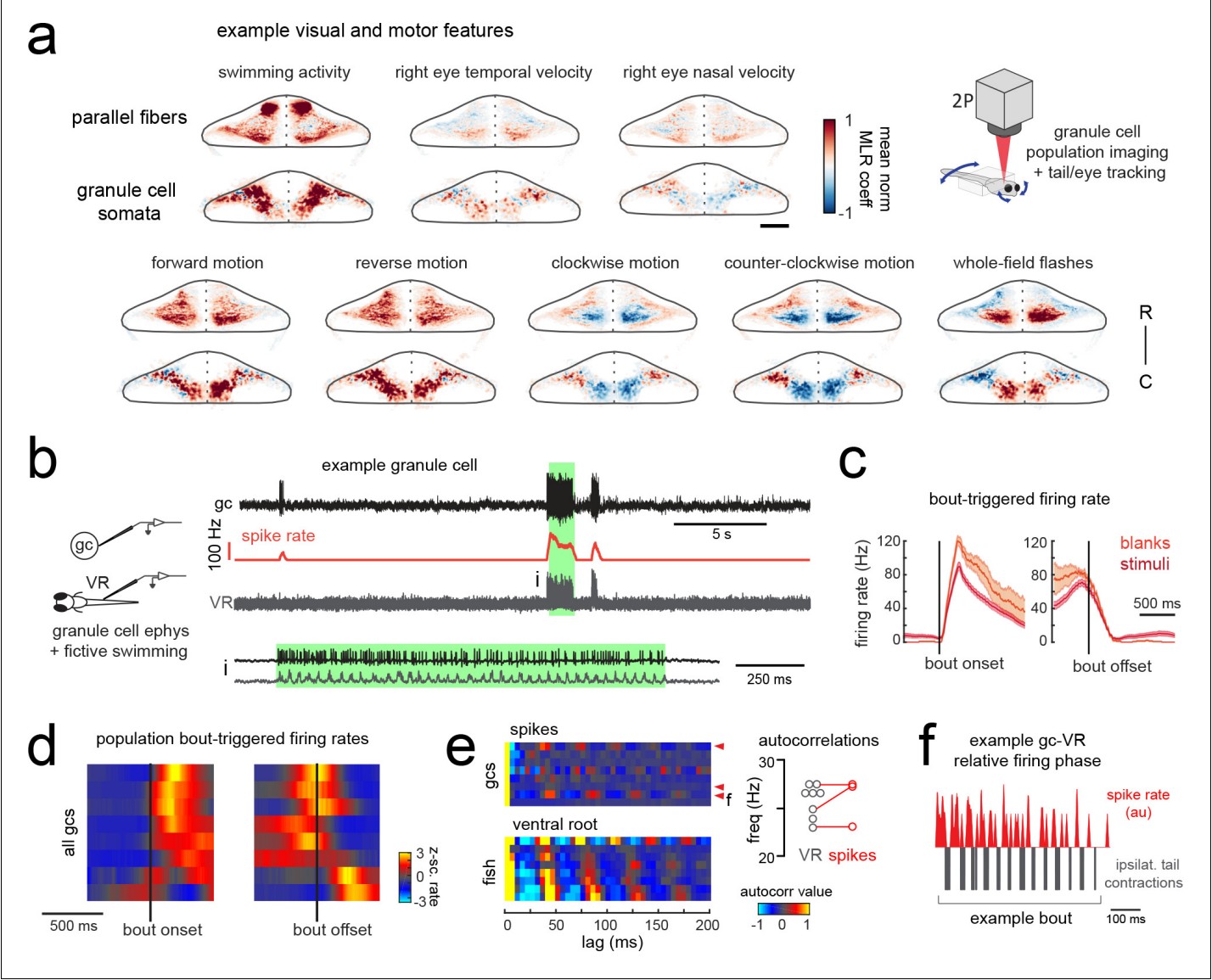

**Figure 6.** Granule cells across the cerebellum code for motor activity with high fidelity. (**a**) Heatmaps of the z-projected mean voxelwise correlation coefficients of two-photon granule cell GCaMP6s signals from multilinear regression with example sensory and motor regressors averaged across seven fish (see Materials and methods). Scale bar = 50 microns. Upper right, cartoon of experimental set-up. (**b**) Left, cartoon of experimental set-up. Right, upper panel, example cell-attached recording from a granule cell (gc, upper trace, black) from a blank trial with simultaneous ventral root recording (VR, lower trace, gray). The granule cell firing rate is also shown (spike rate, middle trace, orange). The bout highlighted in green (**i**) is shown below on an expanded timescale. (**c**) The bout on- (left) and off- (right) triggered mean firing rates for this granule cell during blank recordings (orange) and stimulus trials (red). (**d**) Z-scored heatmaps of bout on- (left) and off- (right) triggered mean firing rates in all granule cells sorted by mean firing rate in the 300 ms following bout onset (N = 8 cells from eight fish). (**e**) Mean autocorrelation heatmap for spikes (upper panel) and ventral root recordings (VR, lower panel) for all granule cells from d), sorted by time to first peak in the VR autocorrelation. The red arrowheads signify granule cells with significant spike autocorrelations during fictive swim bouts (N = 3; p<0.001, Ljung-Box Q-test; see Materials and methods). Right, the first significant peak in the VR autocorrelation for each recording is plotted to give the mean fictive swim frequency for each fish. The red circles are the mean spike autocorrelation frequency obtained from the three significantly autocorrelated granule cells. (**f**) An example bout from the cell indicated in e), which was located ipsilateral to the ventral root recording. The smoothed spike rate (red) is in antiphase with the ipsilateral fictive tail contractions (grey). See also *Figure 6—figure supplement 1*.

DOI: https://doi.org/10.7554/eLife.42138.017

The following figure supplement is available for figure 6:

**Figure supplement 1.** Many granule cells show significant modulation of their firing rates during fictive swimming bouts.

DOI: https://doi.org/10.7554/eLife.42138.018

electrophysiological experiments. The widespread increases in Purkinje cell calcium signals observed in the behaving animal during swimming (*Figure 2a,b*) therefore are likely to reflect simple spike bursts in Purkinje cells (*Figure 1—figure supplement 2*) driven by the high frequency firing of one or more presynaptic granule cells carrying motor efference information.

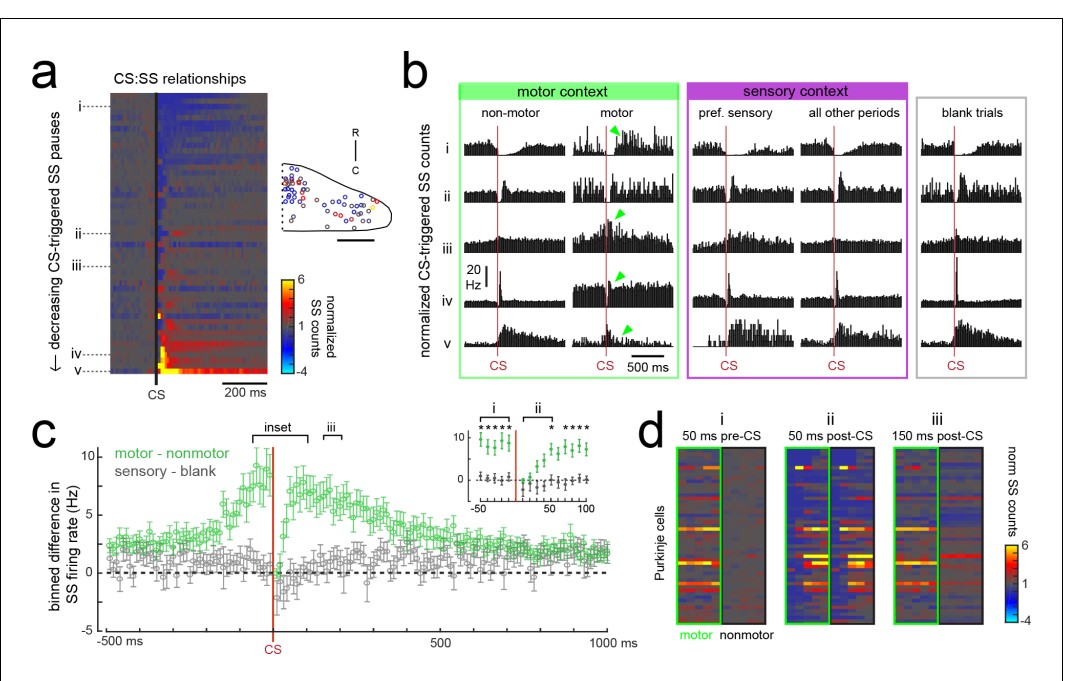

**Figure 7.** Purkinje cells modify their simple spike output in a complex spike- and motor context-dependent way. (**a**) Heatmap of complex spike-driven simple spike (CS:SS) counts for each cell normalized to the mean over 100 ms preceding a complex spike. Cells are sorted by decreasing simple spike pause and increasing excitation. The inset shows the location of these cells colored by the normalized difference in simple spiking in the 50 ms following the CS. (**b**) The mean complex spike-triggered simple spike count (10 ms bins) is shown for five example cells (as indicated in a) for five different contexts. Left (green box), in the presence ('motor') versus absence ('non-motor') of fictive swimming episodes. Under non-motor conditions these different Purkinje cells show, respectively, a CS-induced i) long SS pause, ii) short SS pause with rebound increase, iii) no change in SS, iv) short SS increase, and finally a v) long SS increase. Green arrows highlight changed patterns during motor context. Middle (magenta box), CS:SS relationships across preferred versus all other sensory contexts (only non-motor periods included). Right (grey box), the CS:SS relationship during blank trials (no stimuli, only non-motor periods). Vertical scale bar indicates the rate conversion for 0.2 spikes/10 ms bin (20 Hz). (**c**) Green markers show the mean normalized simple spike rates (calculated from 10 ms bins) for all Purkinje cells centered on the occurrence of a complex spike during a fictive bout minus those occurring at any other point (N = 51 cells). Data are mean ± SEM. Grey markers, simple spike rates centered on the occurrence of a complex spike during all sensory stimuli minus those occurring during blank trials (N = 53 cells). The dashed black line indicates zero difference between conditions. Inset, the window around complex spike onset shown on an expanded timescale. Asterisks indicate p<0.05 for motor minus nonmotor conditions (green markers) as computed by the Wilcoxon signed rank test. Grey markers, no significant differences. (**d**) Heat maps are shown for individual Purkinje cell binned simple spike counts over the three different 50 ms periods as indicated in e). Complex-spike triggered simple spike counts are separated for each cell for those complex spikes occurring during a fictive bout (left column of heatmaps, outlined in green) or at any other time (right column of heatmaps, outlined in black).

DOI: https://doi.org/10.7554/eLife.42138.019

The following figure supplement is available for figure 7:

**Figure supplement 1.** Individual Purkinje cells preferentially combine sensory and motor information.
DOI: https://doi.org/10.7554/eLife.42138.020

## Purkinje cells combine sensory and motor information from distinct inputs

Our data suggests that as a population, Purkinje cells preferentially encode visual features in their complex spike activity whereas swimming activity, arriving in the form of motor efference copies, is predominantly encoded by simple spikes. Breaking this down by group, we find that Purkinje cells belonging to the three different visual complex spike phenotypes described above have simple spike activity that is correlated most strongly with motor activity (fraction of total signal from motor regressors = $0.65 \pm 0.05$, $0.54 \pm 0.07$, $0.76 \pm 0.02$ for the three visual complex spike phenotypes; *Figure 7—figure supplement 1a*). In contrast, for the small group of Purkinje cells with dominant motor-related complex spike phenotypes, the contribution of motor activity to simple spike activity is relatively low ($0.35 \pm 0.12$) and simple spikes are instead broadly influenced by a combination of sensory and motor features (*Figure 7—figure supplement 1a*). This relationship holds true for individual Purkinje cells as well (*Figure 7—figure supplement 1b*). Together, these data suggest that sensory and motor information is preferentially combined in Purkinje cells from distinct sources.

## Motor context alters the relationship between complex spike and simple spike activity

It is well-established that the occurrence of a complex spike can alter simple spike activity in a Purkinje cell both acutely and across longer timescales (*De Zeeuw et al., 2011*). On a short timescale, complex spikes typically cause a brief pause of tens of milliseconds in simple spike firing that can be followed by an increase or decrease in simple spikes lasting hundreds of milliseconds. The particular complex spike-triggered change in simple spiking is robust for a given Purkinje cell but varies considerably across cells (*Zhou et al., 2014*; *Zhou et al., 2015*; *Xiao et al., 2014*). Similar to previous findings, we observed heterogeneity in the relationship between complex and simple spikes across Purkinje cell recordings (*Figure 7a*). At the most extreme end, we observed complex spike-induced pauses or increases in simple spike rates in different cells that took several hundred milliseconds to return to baseline. These pauses or increases in simple spiking may be attributable to a toggling action of the complex spike to shift the Purkinje cell between 'up' and 'down' states (*Loewenstein et al., 2005*; *Sengupta and Thirumalai, 2015*). Several cells had brief pauses (tens of milliseconds) following a complex spike which left simple spikes otherwise unchanged, whereas others showed a brief increase in simple spike firing. Previous studies have suggested that the modulation of simple spike firing by a complex spike is related to the cell's location within the cerebellum (*Zhou et al., 2014*; *Zhou et al., 2015*). We did not, however, observe any clear spatial organization of the complex spike-simple spike relationship in this dataset (*Figure 7a*).

The current behavioral state of the animal should provide important contextual information for cerebellar circuits, therefore we hypothesized that the modulation of simple spikes by a complex spike might be altered in different sensory and motor contexts. In periods during which the fish was at rest (no fictive swimming), the relationship between a complex spike and the simple spike firing rate was similar whether or not visual stimuli were being presented (*Figure 7b*, 'non-motor' versus 'blank trials'). When the fish was performing a fictive swim bout however, the effect of a complex spike on simple spike output appeared diminished (*Figure 7b*, 'non-motor' versus 'motor'), which was not the case for complex spikes occurring during a cell's preferred complex spike sensory stimulus versus those occurring during all other periods (*Figure 7b*, 'pref. sensory' versus 'all other periods').

The unique effect of motor context on this relationship is likely related to the finding that many Purkinje cells have simple spike rates that are strongly excited by motor activity (*Figure 5e*), therefore a complex spike stochastically occurring during a bout would be faced with simple spikes rates that are significantly higher than baseline. Upon closer examination of the temporal window around the occurrence of a complex spike, we observed that the acute effect of a complex spike to modulate simple spike rates was identical between motor and non-motor periods for only a 50 millisecond period following the complex spike, after which time simple spiking returned to high levels correlated with ongoing behavior (*Figure 7c,d*). This temporal window was the same across cells regardless of whether the baseline modulation by a complex spike was to pause or facilitate simple spike firing. These findings suggest that the acute effect of a complex spike to change simple spike output in a Purkinje cell is temporally restricted by the behavioral state of the animal and that plasticity

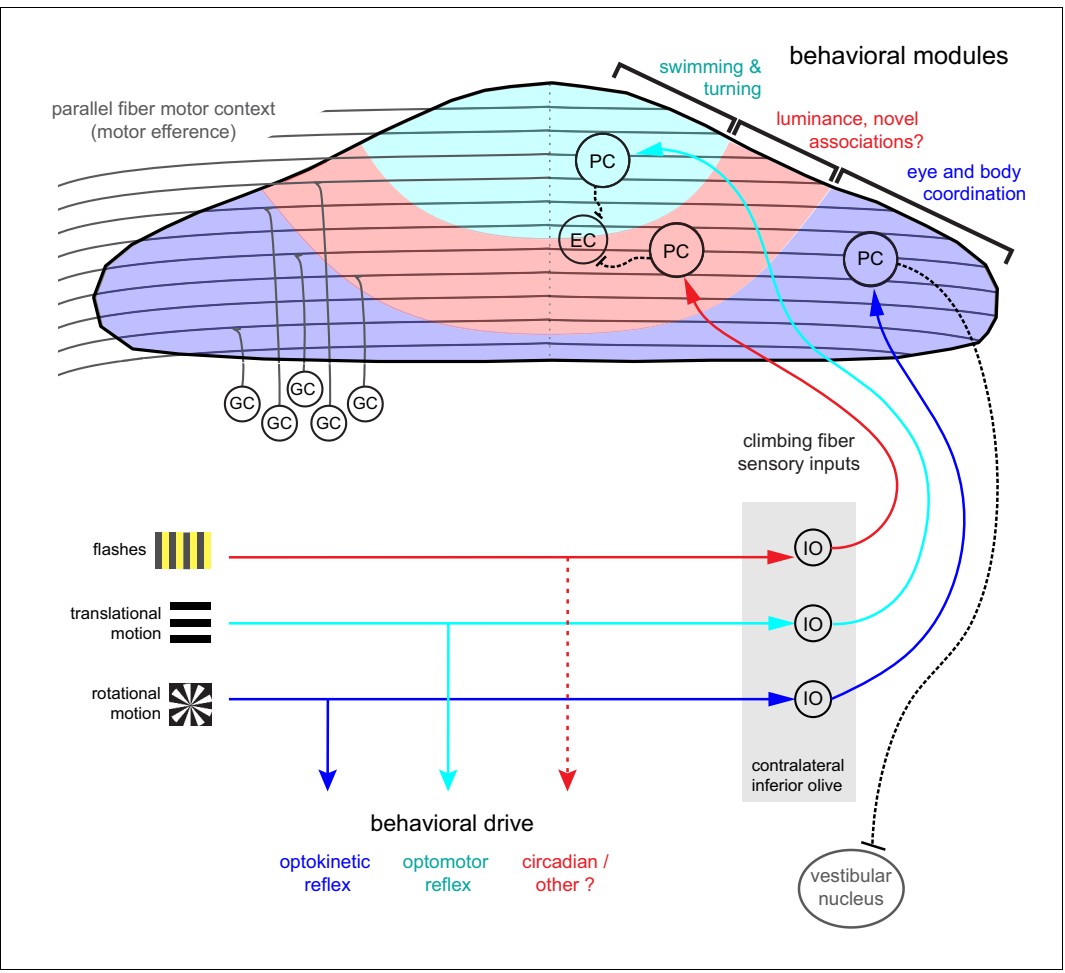

**Figure 8.** Organization of the larval zebrafish cerebellum Granule cells (GCs) send long parallel fibers (grey lines) that contact Purkinje cells (PCs) across the cerebellum and broadly relay motor efference copies of locomotor activity (swimming). Sensory information relating to different visual features are sent by climbing fibers of inferior olive neurons (IO) to stereotyped regions of the contralateral Purkinje cell layer. These visual stimuli contribute to several reflexive behaviors; rotational motion drives the optokinetic reflex of the eyes, translational forward motion drives the optomotor swimming reflex while others, such as luminance, may drive behavior over longer (e.g. circadian) timescales. The three distinct functional regions in the zebrafish cerebellum defined by Purkinje cell complex spike sensory responses that encode these different visual features represent putative behavioral modules. Information about the onset of directional translational motion is preferentially sent to PCs in the rostromedial region of the cerebellum (cyan) and would be important for coordinating turning and swimming, while information about the direction and velocity of rotational motion as would be needed for coordinating eye and body movements is sent to the caudolateral region (blue). The central region (red) receives information about luminance and may provide a substrate for learned sensorimotor associations. Axons from PCs (black dashed lines) of the rostromedial and central regions have mostly internal axons that contact eurydendroid cells (EC) within the cerebellar cortex. Axons from PCs in the caudolateral region have mostly external axons that exit the cerebellum and contact neurons in the caudally-located ipsilateral vestibular nucleus.

DOI: https://doi.org/10.7554/eLife.42138.021

mechanisms relying on coincident complex spike and simple spike activity will have a unique dependence on motor context (see discussion).

## Discussion

In this study, we have taken advantage of an innate set of visually-driven motor behaviors in the larval zebrafish to comprehensively interrogate how Purkinje cells encode sensory and motor features

relating to these behaviors at high spatial and temporal resolution across the cerebellum. Our population imaging data across both the Purkinje cell and granule cell populations are supported by single cell electrophysiological recordings that elucidate complex spike and simple spikes. We furthermore show the robustness and specificity of these patterns across behavioral conditions regardless of whether the tail and/or eyes are freely moving or paralyzed. We show that Purkinje cells fall into anatomically clustered regions that are functionally defined by complex spike responses that convey mostly sensory information. On the other hand, simple spikes convey mostly motor-related information about tail and eye movement. During visuomotor behaviors, these two input streams converge on Purkinje cells in specific regions of the cerebellum and communicate the presence of distinct visual features together with motor context. Each of these regions therefore likely represents a behavioral module whose neural computations are used to guide sensorimotor integration and motor learning in the cerebellum.

## Anatomical and functional organization of cerebellar regions

The three distinct regions formed along the rostrocaudal axis of the larval zebrafish cerebellum we define here based on distinct Purkinje cells sensory complex spike phenotypes to visual stimuli should receive topographically-specific climbing fiber inputs from the inferior olive (*Figure 8*; *Ozol et al., 1999*). The presence of these discrete complex spike response phenotypes across Purkinje cells suggests that zebrafish climbing fiber inputs from the inferior olive have undergone refinement by seven dpf to innervate only one Purkinje cell, in support of other findings (*Hsieh et al., 2014*; *Hsieh and Papazian, 2018*). Ongoing work characterizing the physiology and anatomy of inferior olive neurons and their climbing fibers (unpublished observations) supports this regional characterization and, together with studies of Purkinje cell output to eurydendroid cells, will further our understanding of these anatomical regions.

Differences in developmental timing (e.g. birthdate) are known to contribute to the formation of a topographic functional map in the cerebellum across species (*Hashimoto and Hibi, 2012*). In zebrafish, Purkinje cell development occurs in waves that map onto the same regions we describe here, beginning with a large rostromedial cluster and a smaller, caudolateral cluster and later filling in the central region to form a continuous layer (*Hamling et al., 2015*). Just like in mammals, all climbing fibers cross the midline after leaving the inferior olive and contact the somata or proximal dendrites Purkinje cells in the contralateral hemisphere of the zebrafish cerebellum (*Takeuchi et al., 2015*). The topography of early afferent climbing fiber connectivity onto Purkinje cells is likely hard-wired, as in mammals it is guided by chemical cues and does not depend on developmental activity (see *Apps and Hawkes, 2009* for review). Although all ipsilateral climbing fibers enter the cerebellar cortex as one bundle at larval stages, in the adult, additional fiber bundles are visible (*Takeuchi et al., 2015*), suggesting that other routes or types of information are added for communication between the inferior olive and cerebellar cortex at later stages. Regional differences in cytoarchitecture and patterns of molecular markers such as zebrin have also been useful for identifying related Purkinje cells into groups in the mammalian cerebellum (see *Cerminara et al., 2015* for review). Although in larval zebrafish all Purkinje cells are zebrin-positive (*Bae et al., 2009*), many other genes are expressed in restricted patterns in the zebrafish (*Takeuchi et al., 2017*) and mammalian cerebellum (*Hawkes, 2014*) that may help define the subdivision of Purkinje cells into clearly-defined subregions within the cerebellum.

The organization of the cerebellum is thought to impart distinct functional roles across regions, such that each group of Purkinje cells processes sensorimotor information relating to a different behavioral component (*Witter and De Zeeuw, 2015*). Although we only probed one sensory modality to drive behavior, zebrafish are highly visual animals that perform robust visuomotor behaviors at the larval stage, including prey capture, optokinetic and optomotor responses, and associative learning with a conditioned visual stimulus (*Easter and Nicola, 1996*; *Budick and O'Malley, 2000*; *Aizenberg and Schuman, 2011*; *Harmon et al., 2017*). Visual information is therefore a highly salient sensory modality at this age and in accordance with this strong ethological relevance we find that the complex spike sensory responses to visual stimuli provide an overarching organization of the Purkinje cell layer into putative behavioral modules (*Figure 8*). Previous studies have used confocal imaging and optogenetics to identify general regions of the cerebellum that are important for optomotor and optokinetic responses (*Matsui et al., 2014*). Our current results build on these maps with an expanded set of visual stimuli, high-resolution two-photon population imaging, and single-

cell electrophysiology, to comprehensively describe visual and motor feature coding from the level of single spikes to population activity.

We see little evidence for the encoding of multiple visual features in these complex spike responses, however we expect to find representations of features from multiple sensory modalities in individual Purkinje cells arising from the multimodality of inferior olive neurons (*Ohmae and Medina, 2015*; *Ju et al., 2018*). It will be of great interest to see if the same spatial mapping by complex spike phenotype is conserved across other sensory modalities. Many other sensory systems are active at this age and provide salient stimuli for larval zebrafish as demonstrated in behavioral studies. Larval zebrafish show innate behavioral startle responses to loud auditory cues across a range of frequencies (*Bhandiwad et al., 2013*), however functional imaging across the brain suggests that the neural coding of auditory stimuli at this stage is generic and underdeveloped in zebrafish compared to the visual modality (*Vanwalleghem et al., 2017*). In contrast, the activity of many neurons across the brain including the cerebellum are differentially modulated by vestibular inputs at larval stages (*Favre-Bulle et al., 2018*; *Migault et al., 2018*). Given the pronounced complex spike responses we observe in response to rotational motion in the Purkinje cells with axonal output to the vestibular nucleus, it is likely that the coordination of vestibular and visual inputs during rolling movements (the vestibulo-ocular reflex), critically engages the cerebellum of the larval zebrafish. Although the larval zebrafish exhibits a broad repertoire of innate behavioral responses to many other stimulus modalities including touch, the lateral line, and olfaction (see *Fero et al., 2011* for review), there is a lack of physiological data to understand how these signals are encoded in the central nervous system and the cerebellum in particular. It is furthermore conceivable that other sensory systems become more important at later developmental stages, for examples olfactory processing of cues for kin recognition and social behaviors in the juvenile fish (*Dreosti et al., 2015*) or lateral line-mediated schooling behaviors in the adult (*Miller and Gerlai, 2012*).

In support of the spatial division of the cerebellum by visual complex spike phenotypes that we show in the current study, previous findings from a completely different behavioral paradigm in larval zebrafish also found three complex spike regions with a similar organization. *Harmon et al., 2017* developed an associative learning task to pair visual stimuli with an electric shock to elicit conditioned swimming responses. They found, using single-cell electrophysiological recordings, that Purkinje cell complex spike patterns in their conditioning paradigm were spatially and functionally clustered into three regions along the rostrocaudal axis that overlap well with the regions described here. These complementary findings suggest that this regionalization is of fundamental importance across modalities and behaviors. However, experimental attempts at associative learning using auditory stimuli have been unsuccessful at this stage and even in the 6 week-old larva (*Thompson, 2016*), suggesting a prioritization of visual information for the earliest motor learning in larval zebrafish. Future studies are needed to examine how the function and organization of the zebrafish cerebellum across regions may change to reflect an increasing complexity and repertoire of sensorimotor behaviors at later developmental stages.

## Complex spikes use different temporal bases to encode specific visual features important for the animal's behavioral repertoire

Our results show that that majority of Purkinje cells across the cerebellum encode visual and not motor information in their complex spike activity. We observed a remarkably discrete and complete classification of nearly all Purkinje cells (>90%) for a specific visual complex spike phenotype whose sensory nature was clearly distinguishable from motor-related signals of eye and tail behavior. These visual complex spike phenotypes were distinct between the three groups of Purkinje cells in different rostrocaudal regions. Below, we discuss how the representation of these different visual features may serve as behavioral modules that relate to the particular behavioral repertoire of the larval zebrafish and to findings from the mammalian literature.

Transient changes in the direction of translational visual motion convey information critical for driving locomotion and turning behaviors, or in the case of visual reafference, for evaluating the success of a directed behavior. In the larval zebrafish, Purkinje cells of the rostromedial cerebellum reliably encode acute, directional changes of motion in the visual field with a preferred directional tuning. During the optomotor response, fish swim to stabilize their position with respect to the visual field. Larval zebrafish also perform a variety of low and high-angle turns at this stage while exploring, performing escape maneuvers, and hunting prey, therefore complex spike signals updating the brain

about a transient change in motion in the visual field have strong ethological relevance. This population of Purkinje cells whose complex spikes encode directionally-selective motion onset are reminiscent of the directionally-tuned Purkinje cells in the oculomotor vermis of posterior lobes VI and VII in primates, where complex spike tuning organizes the cells into functional groups whose simple spikes encode real-time eye motion (*Soetedjo and Fuchs, 2006*; *Herzfeld et al., 2015*).

In the caudolateral region of the cerebellum we find Purkinje cells with strong complex spike responses to unidirectional rotational motion and axons that project primarily to the octaval (vestibular) nuclei in zebrafish (*Matsui et al., 2014*). These Purkinje cells fire complex spikes during visual motion in a temporal to nasal direction presented to the ipsilateral eye (with respect to the anatomical location of the Purkinje cell), resulting in a tonically elevated complex spike rate during visual motion in the preferred direction. This caudal region is likely the vestibulocerebellum, homologous to the mammalian flocculonodular lobe where Purkinje cells receive climbing fiber input conveying information about ongoing, opposing directional visual and rotational head motion that is used for vestibulo-ocular coordination (*Simpson and Alley, 1974*; *Ito, 1982*). Complementary imaging studies in larval zebrafish show strong, directionally-tuned responses in the activity of undefined cerebellar neurons in this same region (*Favre-Bulle et al., 2018*; *Migault et al., 2018*). Larval zebrafish perform slow steering maneuvers of the tail while navigating and also produce smooth eye movements while engaging in activities such as prey tracking (*McElligott and O'malley, 2005*), both of which result in slow changes in the visual field. In addition, zebrafish can control their eyes independently from each other, so these signals are likely to be integrated in the brain together with vestibular and body axis information to achieve coordinated movements. Notably, these zebrafish vestibulocerebellar Purkinje cells show complex spike responses not only to rotational but also to translational moving fields, which is not seen in mammals but has been observed in pigeons (*Wylie and Frost, 1991*). This finding may relate to the additional complexity of optic flow that arises during navigation in a three-dimensional world for birds and fish.

We furthermore observed that Purkinje cells in the caudolateral region have spontaneous complex spike rates an order of magnitude higher than those in the rostromedial region described above and show sustained high complex spike firing in response to their preferred stimulus. This could allow for increased temporal precision in order to generate fast and precise firing patterns as would be required when generating sensorimotor associations or coordinating smooth movements (*Porrill et al., 2013*; *Suvrathan et al., 2016*). The computation itself may in fact be different in this region since complex spikes could use conventional rate coding to encode the speed and direction of ongoing, slow movements of the visual field during behavior, as proposed by *Simpson et al., 1996* based on observations in the mammalian flocculonodular lobe across species. These findings challenge the assumption that the computations being performed across the cerebellum all follow the same rules and that the occurrence of a discrete event, rather than information about an ongoing event, is transmitted by complex spikes.

The heterogeneity of sensory complex spike coding of luminance in the intermediate region of the zebrafish cerebellum sets this group of Purkinje cells apart from the other two visual phenotypes We see both many differences in responses to luminance changes, including light/dark preference, tonic/phasic responses, latency from stimulus onset to complex spike, and receptive field size. We propose that these Purkinje cells are therefore well-suited to modulate a diversity of light-mediated behaviors in the larval zebrafish. Although the luminance stimuli in the current experiments were titrated to be moderate and thus not evoke acute behavioral responses, sudden strong decreases in luminance induce re-orienting navigational turns (*Burgess and Granato, 2007*) or escapes (*Temizer et al., 2015*) in zebrafish larvae and transient startle responses in the adult (*Easter and Nicola, 1996*), the latter two representing likely predator avoidance responses. With respect to these fast behaviors, a transient encoding of luminance change could serve to modulate these response circuits. Luminance increases spontaneous locomotor activity in larval zebrafish over longer timescales as well, which is used as a cue to regulate circadian rhythms and motivate feeding and exploratory behavior in the daytime (*Burgess and Granato, 2007*). There is also an innate preference for larval zebrafish to be in lighter areas of their environment, a behavior known as phototaxis (*Brockerhoff et al., 1995*). These latter behaviors would more likely make use of rate coding of ambient luminance, as observed in the complex spike output of some Purkinje cells, to provide sensory integration over long timescales (tens of minutes).

The differing luminance preferences and temporal dynamics across this group may furthermore be useful for learning novel associations. Indeed, a recent report by *Harmon et al., 2017* found that Purkinje cells in this central area of the larval zebrafish cerebellum (termed 'multiple complex spike cells' in this study) preferentially acquired complex spike responses to a conditioned visual stimulus during associative learning. As mentioned above, Purkinje cells in this central region are also born slightly later in development compared to the groups described above (*Hamling et al., 2015*), findings that together suggest this region may preferentially contribute to flexible or learned sensorimotor behaviors. This region may be similar to areas in the central zone (posterior lobes VI and VII) of the cerebellum in mammals, which support a wide range of behavioral functions (*Koziol et al., 2014*; *Stoodley et al., 2012*).

## What signals are complex spikes encoding?

There is great debate about whether climbing fiber signals convey error, predictive, or novelty signals (see *Simpson et al., 1996* and *Streng et al., 2018* for reviews). The error hypothesis would suggest that the visually-evoked responses we observe here signal unexpected events or 'negative sensory events to be avoided' such as retinal slip (*Lang et al., 2017*). However, these signals are not necessarily a classical error signal (*Ito, 2013*), because in the current study we find that stimulus-evoked complex spikes are equally prominent in paralyzed fish as in experiments where the eyes and tail are free and track the stimulus. Furthermore, many complex spikes are robustly elicited by visual stimuli that do not acutely drive behavior, such as reverse motion or luminance changes.

Other work suggests that climbing fibers carry instructional signals for upcoming motor actions in a learned behavior, in the context of a reinforcement learning signal (*Ohmae and Medina, 2015*; *ten Brinke et al., 2015*; *Heffley et al., 2018*), or could provide the corrective drive used to initiate locomotion (*Ozden et al., 2012*). While we are not testing predictive signals in this study, it is nonetheless clear that for the complex spikes elicited during visual stimuli in our experiments do not signal an upcoming motor event. In cases when complex spikes are driven by the onset of directional translational motion, they occur with a consistently short latency (approximately 200 ms) whereas the latency to swim bout onset is much longer and more variable, and the presence of absence of these visual complex spikes across trials do not predict the occurrence of a swim bout.

Other hypotheses suggest that climbing fibers may encode novelty or salience signals related to sensory stimuli, although in fact these hypotheses do not exclude the previous ones since climbing fibers may be able to carry different types of signals by multiplexing (*Ohmae and Medina, 2015*). The complex spike responses we observe in this study do not encode all novel or salient visual stimuli as we see that responses are selective for certain visual features. In our experiments, complex spikes do not habituate but are consistently elicited by visual stimuli, across many trials and many hours, in contrast to what might be expected if complex spikes encoded novelty. It remains however to be seen how robust these responses are over longer timescales, as previous work has suggested the complex spike response to a novel sensory stimulus is subject to habituation only with repeated exposure across many days (*Ohmae and Medina, 2015*).

Since the above hypotheses were mostly developed with respect to observations in the context of cerebellar learning, the role of complex spikes may be different for innate feature coding. Our results suggest an innate coding of sensory features in climbing fiber signals in the naïve animal, consistent with observations of visual and multimodal sensory responses carried by climbing fibers in other studies in zebrafish (*Hsieh et al., 2014*; *Sengupta and Thirumalai, 2015*; *Scalise et al., 2016*; *Harmon et al., 2017*) and mammals (*Ohmae and Medina, 2015*; *Ju et al., 2018*). The complex spikes resulting from climbing fibers tuned to specific sensory features could subsequently drive the learning that underlies novel associations, including predictions, as arises when an animal experiences the repeated pairing of a complex spike-evoking stimulus and a motor event (*Ito, 2001*; *Harmon et al., 2017*). In this context, the sensory complex spike signal could be interpreted as a sensory prediction error that drives associative learning. Additional work is needed to determine how the complex spike responses encode different sensory modalities both in the naïve animal and throughout the course of learning.

## Motor efference copies in the cerebellum

Our population-wide imaging and extensive electrophysiological recordings show that most Purkinje cells across the cerebellum encode the current behavioral state (motor context) of the animal through a pronounced increase in simple spikes during locomotor behaviors. We observed strong swim-related signals in granule cells and Purkinje cells during both active and fictive swimming, where zebrafish were awake but paralyzed, therefore this activity is more consistent with motor efference copy signals than proprioceptive or lateral line activation. Moreover, we found that simple spike output correlated best with the strength of ongoing swimming rather than reporting a phasic or binary locomotor state, supporting previous findings that motor parameters are linearly coded in the cerebellum (*Raymond and Medina, 2018*). Only a small minority of Purkinje cells showed a motor-related decrease in simple spiking, which may reflect the relatively small contribution of feed-forward inhibition via molecular layer interneurons. The increases in simple spike output that we observe are far less heterogeneous than the effects of locomotion on mammalian Purkinje cell simple spike firing (*Jelitai et al., 2016*; *Sauerbrei et al., 2015*) and build on previous electrophysiological samplings of Purkinje cell activity that showed increases in membrane depolarization and simple spike output during fictive swimming in zebrafish (*Sengupta and Thirumalai, 2015*; *Scalise et al., 2016*).

These results suggest that motor efference signals during whole-body locomotion (swimming) drive simple spike output in nearly all cerebellar Purkinje cells in the larval zebrafish. Our current granule cell population imaging and electrophysiological recordings in zebrafish together with other recordings and optogenetic experiments in zebrafish and mice (*Ozden et al., 2012*; *Powell et al., 2015*; *Jelitai et al., 2016*; *Giovannucci et al., 2017*; *Knogler et al., 2017*; *Albergaria et al., 2018*) provide strong evidence that the cerebellum broadly encodes intended locomotor output or signals related to it in the input layer (*Figure 6*). These findings suggest an enrichment of motor signals across parallel fiber inputs, though some regional specialization of signals in limbed vertebrates may be needed to coordinate different limb networks. Future work is required to investigate the origin of mossy fibers carrying eye and tail motor efference copies to the zebrafish cerebellum and how these signals are transformed by subsequent processing stages in cerebellar circuits.

## Complex spike - simple spike relationships

We observed that the dominant action of motor activity on simple spiking acutely changes how complex spikes and simple spikes interact in Purkinje cells. During non-locomotor periods, complex spikes have the ability to consistently increase or pause simple spiking for several hundreds of milliseconds in different Purkinje cells. Under motor-driven conditions of high simple spike rates, however, a complex spike resets simple spike activity for only a brief (<50 ms) window in all Purkinje cells before simple spikes return to their previous high rate. This is likely due to the overwhelming excitatory influence of locomotor activity carried by parallel fibers that drives simple spiking across the Purkinje cell population at high rates. When faced with these high simple spike rates, a complex spike arriving during motor activity therefore has a limited influence over simple spike output. The narrowing of this temporal window may serve to make finer adjustments of motor activity through very acute perturbations in network activity.

Across longer timescales, sensorimotor behaviors needs to be adjusted during development, experience, and learning, so that an animal can adapt to suit a changing environment or context. In a developmental context, the cerebellum may be actively engaged in refining and maintaining sensorimotor behaviors as the physiology of neural circuits, muscles, and sensory appendages matures. In the context of supervised cerebellar learning, classical theories predict that the coincident activation of a climbing fiber input and parallel fiber synapses drives long-term depression at the active parallel fiber to Purkinje cell synapses, leading to motor learning (*Ito, 2001*; but see *Bouvier et al., 2018*). Synaptic plasticity mechanisms both at other synapses and involving other cerebellar neurons (e.g. interneurons) are also likely to contribute (see *Gao et al., 2012* for review). We propose that motor efference signals during swimming and eye movements are widely broadcasted across the cerebellum to Purkinje cells because these are the most relevant signals not only for coordinating ongoing behaviors but also for driving plasticity. The enrichment of motor-related activity across the granule cell layer and subsequent broad excitation of Purkinje cells would support learned associations between motor behaviors and any relevant sensory information carried by regionally-specially

climbing fiber input. Indeed, recent work by *Albergaria et al., 2018* supports this idea by showing that a generalized increase in granule cell excitation during either locomotion or optogenetic stimulation enhances cerebellar learning in a paradigm for eyeblink conditioning. The amenability of the zebrafish cerebellum to in vivo physiological and behavioral recordings together with the hypotheses raised by this study should make it an attractive system to study the rules of cerebellar plasticity and learning in the future.

## Outlook

Our results reveal a strong spatial organization of visual feature encoding in the Purkinje cell population into three rostrocaudal functional regions receiving different climbing fiber inputs. These regions are each involved in processing visual information relating to distinct motor behaviors and as such exhibit unique temporal features in sensory coding. Broad excitation from granule cells is layered on these regions during locomotor activity as a contextual signal. We believe that the system of granule cells and Purkinje cells together thus forms the substrate for cerebellar modules modulating innate and learned motor behaviors. These and other recent findings (*Matsui et al., 2014*; *Harmon et al., 2017*) provide a promising outlook for using the zebrafish as a model organism for understanding motor control and learning in the cerebellum.

## Materials and methods

### Experimental model and subject details

Zebrafish (*Danio rerio*) were maintained at 28 ˚C on a 14 hr light/10 hr dark cycle using standard protocols. All animal procedures were performed in accordance with approved protocols set by the Max Planck Society and the Regierung von Oberbayern (TVA 55-2-1-54-2532-82-2016). All experiments were performed using larvae at 6–8 dpf of as yet undetermined sex.

To label Purkinje cells specifically, we made use of the *aldoca* promoter and the carbonic anhydrase 8 (ca8) enhancer element as published previously (*Takeuchi et al., 2015*; *Matsui et al., 2014*). For electrophysiological recordings in Purkinje cells, *aldoca*:GFF;mn7GFF;UAS:GFP fish were used (*Takeuchi et al., 2015*; *Asakawa et al., 2008*; *Asakawa et al., 2013*), with Tg(gSAIzGFFM765B); UAS:GFP and Tg(gSAG6A); UAS:GFP fish additionally used for granule cell recordings (*Takeuchi et al., 2015*). For calcium imaging experiments with granule cells, Tg(gSA2AzGFF152B); UAS:GCaMP6s fish were used (*Takeuchi et al., 2015*; *Thiele et al., 2014*). For calcium imaging experiments in Purkinje cells, we cloned GCaMP6s (*Chen et al., 2013*) downstream of the *ca8* enhancer with an E1b minimal promoter referred hereafter as PC:GCaMP6s. We injected PC: GCaMP6s together with *tol2* mRNA in one cell stage embryos (25 ng/µl each), screened at six dpf for expression in the cerebellum, and raised strong positive fish to adulthood. Positive F1 progeny were used for all imaging experiments. For simultaneous electrophysiological and imaging experiments, we injected PC:GCaMP6s without *tol2* mRNA to achieve sparse, single-cell labelling. For anatomical experiments, we created a construct harboring a bright GFP variant mClover3 (*Bajar et al., 2016*) tagged with a membrane targeting signal (Fyn). This construct is termed PC:Fyn-mClover3. Injections were done as described for sparse GCaMP6s labelling in fish expressing *aldoca*:gap43-mCherry to allow registration across fish. For Purkinje cell counting, we created a stable transgenic line as described above where a nuclear localization signal (NLS) is fused to the N-terminus of GCaMP6s (PC:NLS-GCaMP6s) to restrict GCaMP6s to the nucleus.

### Visual stimuli

For functional imaging experiments, trials were presented that consisted of the following stimuli, in non-randomized order: Black and white whole-field gratings were presented with motion in the forward direction at slow, medium, and fast speeds (3, 10, and 30 mm/s, respectively), for five seconds each with a pause of five seconds between stimuli, followed by reverse, leftward, and rightward moving gratings of the same duration and at medium speed. Grating remained static between stimuli. Black and white windmill patterns were rotated at 0.2 Hz with changing velocity that followed a sine function. Windmill patterns were presented across the whole field as well as for each half of the visual field. Flashes covered the whole visual field and switched between maximum luminance and darkness. For electrophysiological recordings, stimuli were similar as for functional imaging with the

exception that the stimulus set had shorter pauses between stimuli and that fewer repetitions of the rotating windmill stimulus were presented. Blank trials consisting of static gratings were also interspersed with stimuli trials to obtain baseline responses. For one experiment (*Figure 4—figure supplement 1g*) the fish was also presented with a series of whole-field black or white flashes of various durations (50–5000 ms) against a baseline intermediate luminance.

## Functional population imaging

Volumetric functional imaging in the larval zebrafish cerebellum was performed as previously described in *Knogler et al., 2017*. Briefly. 6–8 dpf nacre (*mitfa -/-*) transgenic zebrafish larvae with GCaMP6s expressed in Purkinje cells were embedded in 1.5–2.5% agarose prior to imaging. Neural activity was recorded with a custom-built two-photon microscope. A Ti- Sapphire laser (Spectra Physics Mai Tai) tuned to 905 nm was used for excitation. Larval brains were systematically imaged while presenting visual stimuli (see below) at 60 frames per second using a Telefunken microprojector controlled by custom Python software and filtered (Kodak Wratten No.25) to allow for simultaneous imaging and visual stimulation. We acquired the total cerebellar volume by sampling each plane at ~5 Hz. After all stimuli were shown in one plane, the focal plane was shifted ventrally by 1 µm and the process was repeated. Tail and eye movement was tracked throughout with 850 nm infrared illumination and customized, automated tracking software. Behavior was imaged at up to 200 frames per second using an infrared-sensitive charge-coupled device camera (Pike F032B, Allied Vision Technologies) and custom written software in Python.

## Image processing

Image analysis was performed with MATLAB (MathWorks) and Python similar to *Knogler et al., 2017*. Python analysis used scikit-learn and scikit-image (*Pedregosa et al., 2012*; *van der Walt et al., 2014*). Volumetrically-acquired two-photon data was aligned first within a plane then across planes to ensure that stacks were aligned to each other with subpixel precision. Any experiments during which the fish drifted significantly in z were stopped and the data discarded. The boundary of the cerebellum was manually masked to remove external signals such as skin autofluoresence. All signals from all planes were extracted for voxelwise analysis (mean of approximately 350 billion $\pm$ 10 billion for 5 fish with 100 planes with an additional 118 billion for a sixth fish with only 34 planes). Purkinje cell ROI activity traces were extracted using automated algorithms based on local signal correlations between pixels (see *Portugues et al., 2014* for details) and used for principal component analysis (see Materials and methods below). Tail activity during imaging experiments was processed to yield a vigor measurement (standard deviation of a 50 ms rolling buffer of the tail trace) that was greater than zero when the fish is moving. Independent left and right eye position and velocity were obtained from eye tracking data.

## Single cell Purkinje cell imaging

Sparse labelled Purkinje cells expressing GCaMP6s were used to perform two-photon imaging as described above to identify any signal compartmentalization (*Figure 1—figure supplement 2*). Visual stimuli consisting of reverse and forward moving gratings were probed to evoke signals in Purkinje cells. For five Purkinje cells across three fish, ROIs for soma and parts of the dendrite were drawn manually and Calcium traces were extracted using custom-written software in Python. The most distal dendritic ROI was correlated with somatic ROI to determine the correlation coefficient for each cell.

## Electrophysiological neural recordings

Cell-attached electrophysiological recordings were performed in 6–8 dpf zebrafish as previously described (*Knogler et al., 2017*) using an Axopatch Multiclamp 700B amplifier, a Digidata series 1550 Digitizer, and pClamp nine software (Axon Instruments, Molecular Devices). Data were acquired at 8.3 kHz using Clampex 10.2. Wild-type or transgenic zebrafish larvae with GFP-positive Purkinje cells and motor neurons were used for most recordings (see subject details above).

Larvae were paralyzed in bath-applied buffered 1 mg/ml alpha-bungarotoxin (Cayman Scientific, Concord, CA) and embedded in 1.5% low melting point agarose in a 35 mm petri dish. External solution was composed of Evans solution (134 mM NaCl, 2.9 mM KCl, 2.1 mM CaCl2, 1.2 mM

MgCl2, 10 mM glucose, 10 mM HEPES, pH 7.8 with NaOH). Electrodes for neuron recordings (6–12 MΩ) were pulled from thick-walled borosilicate glass with filament and were filled with the following intracellular solution (in mM): 105 D-gluconic acid, 16 KCl, 2 MgCl2, 10 HEPES, and 10 EGTA, adjusted to pH 7.2, 290 mOsm (*Drapeau et al., 1999*). Sulforhodamine B (0.1%) was included in the intracellular solution to visualize the electrode. The skin overlying the cerebellum was carefully removed with a glass electrode prior to recording. Post-recording fluorescent images of GFP-positive Purkinje cells and the recording electrode (visualized with an RFP filter) as well as bright-field images to confirm cell identity and map somatic location were acquired with an epifluorescent Thor-Labs camera controlled by Micromanager.

Electrophysiological data were analyzed offline with Clampfit 10.2 software (Molecular Devices) and Matlab (Mathworks, Natick MA). Cell-attached traces were high-pass filtered at 1–10 Hz and complex spikes and simple spike were automatically extracted by setting a threshold for each type of spike in that recording. A 2.5 ms period was blanked following each complex spike so that the complex spike waveform did not cross the simple spike threshold. Baseline firing rates were calculated from blank trials where no visual stimuli were presented or from the two second period at the beginning of each trial prior to the first stimulus onset if no blanks were obtained.

For experiments with simultaneous calcium imaging, stochastically-labeled single Purkinje cells expressing GCaMP6s were recorded with an epifluorescent backlit-CMOS camera (Photometrix Prime 95B) at 11.5 fps controlled by Micromanager and triggered by pClamp software during electrophysiological recordings. No visual stimuli were shown in these experiments. Fluorescent Purkinje cell activity was processed by manual ROI extraction. Extracted complex spike and simple spike rates from simultaneous electrophysiology traces were convolved with a GCaMP6 kernel for comparison with the fluorescent signal.

For electrophysiological recordings in the semi-paralyzed animal, larval zebrafish were embedded in 2% low-melting point agarose and injected with 0.5 mg/ml alpha-bungarotoxin in the caudal tip of the tail. This method reduces the trunk contractions during swimming but preserved full eye movement. The agarose around the eyes was removed and the fish was lit from below with 850 nm infrared illumination to allow for good image contrast of the eyes. Eye movement was recorded during simultaneous electrophysiological recordings and tracked offline with customized, automated software to extract independent trajectories for each eye.

## Ventral root recordings

To obtain extracellular ventral root recordings, a thin-walled borosilicate glass electrode with a large opening (approximately a quarter of the width of a somite) was first used to remove a small section of skin overlying the horizontal myotomes of the spinal cord around the fifth spinal somite. The electrode was then cleared with positive pressure and positioned over the terminals of the ventral root with gentle suction to ensure good signal to noise.

Motor activity was extracted as a moving standard deviation of the ventral root trace. A threshold was then applied to identify ventral root activity that would correspond to motor output on the side of the animal ipsilateral to the recording electrode. To extract a binarized trace of swimming bouts, ventral root activity separated by an interval of less than 100 ms was considered to be part of the same bout. The vigor trace was median filtered to extrapolate vigor information across the entire bout. Peaks in the lag of the autocorrelation analysis of the thresholded, binarized signal was used to extract fictive swim frequency.

## Multilinear regression

Briefly, this analysis involves three steps. First, we processed and extracted physiological signals from imaging data and electrophysiological recordings (see above). Second, we used each different feature of the visual stimulus or motor behavior, such as rotational clockwise visual motion velocity, or the strength of the swimming bouts across a trial, to build a vector of values for each trial convolved with the temporal dynamics of the signal (calcium signal or firing rate). These feature vectors are termed regressors. Third, we performed multilinear regression to quantify the contribution of these different features to the signal of interest. This step included parameter validation to ensure that the results of the analysis are robust. Following this process, each signal is assigned a vector of

coefficient weights that can be multiplied by the set of regressors to best recapitulate the activity of that signal.

Motor regressors were computed for each trial from the behavioral parameters obtained from eye and tail motor information in imaging and electrophysiology experiments (see above). Motor regressors for swimming were created to capture various features including bout onset, offset, duration, and vigor. Eye motor regressors captured directional velocity of each eye independently. Sensory regressors for each type of experiment were the same for all cells and were created using features including the duration, direction, and velocity of moving stimuli as well as luminance (see *Figure 1—figure supplement 2* and *Figure 3—figure supplement 1* for full regressor lists for imaging and electrophysiology).

For functional imaging data, regressors were convolved with a GCaMP6s kernel, modeled as a single exponential function with time decay constant tau = 1600 ms. The tau for this kernel was derived from the average single exponential fit of the fluorescence peak produced by a single complex spike as ascertained by simultaneously recorded GCaMP6s and electrophysiological signals (*Figure 1—figure supplement 2*, N = 8 cells). Regressors were normalized and passed to the scikit-learn function LinearRegression to compute the mulilinear regression coefficients, which was sufficient to accurately recapitulate the calcium traces (mean coefficient of multiple determination = 0.46 ± 0.02).

The higher sampling rates of electrophysiological recordings (8.3 KHz) allowed us to create additional regressors that captured more subtle features in the visual stimuli, for example the onset of translational motion in a given direction. The window for these regressors spanned a 500 ms period beginning at stimulus onset. Our previous electrophysiological recordings in granule cells have suggested a latency of ~100–200 ms for visual input to arrive at the input layer of the cerebellum (*Knogler et al., 2017*) similar to the mean latency of 126 ms reported for visual responses in the mouse inferior olive (*Ju et al., 2018*). Since most sensory stimuli were presented for longer periods (gratings for 5 s, windmill stimuli for 10 s, flashes for 1 s), this short window was designed to be sufficiently long to capture onset-related signals that face synaptic delays, but also clearly distinguish between responses that are transient at stimulus onset or last for the duration of the stimulus. Windmill stimuli had sinusoidal velocity and smoothly changed direction, thus multiple regressors were built for these stimuli that represented graded velocity, binary motion in a given direction, as well as change of direction. These motor and visual regressors were then convolved with a 20 ms filter to match the convolution of spiking into firing rates.

In order to best analyze our electrophysiological data with this extended set of regressors, we implemented a variant of lasso regression known as elastic net regularization using the function *lasso* from MATLAB. This is a useful fitting method for linear regression using generalized penalties that has been shown to be robust and gives sparse coefficient weight distributions such that in practice many regressor coefficients are zero (*Zou and Hastie, 2005*; *Tibshirani, 2011*; *Dean et al., 2015*).

Documentation from MATLAB (r2018b) gives the following formulation:

'Elastic net solves the problem

$$\min_{\beta_0, \beta} \left( \frac{1}{2N} \sum_{i=1}^{N} \left( y_i - \beta_0 - x_i^T \beta \right)^2 + \lambda P_a(\beta) \right), \text{ where}$$

$$P_a(\beta) = \frac{(1-\alpha)}{2} \beta_2^2 + \alpha \beta_1 = \sum_{j=1}^{p} \left( \frac{(1-\alpha)}{2} \beta_j^2 + \alpha \beta_j \right).$$

- $N$ is the number of observations.
- $y_i$ is the response at observation $i$.
- $x_i$ is data, a vector of length $p$ at observation $i$.
- $\lambda$ is a nonnegative regularization parameter corresponding to one value of Lambda.
- The parameters $\beta_0$ and $\beta$ are a scalar and a vector of length $p$, respectively.
- The penalty term $P_a(\beta)$ interpolates between the $L^1$ norm of $\beta$ and the squared $L^2$ norm of $\beta$.'

Alpha values of 0.2 were used which represent an elastic net optimization with only modest sparsification, approaching ridge regression. Increasing the alpha parameter to move closer to an elastic net optimization did not significantly alter the main regressor weights. As the regularization coefficient Lambda increases, the number of nonzero components for regressor weights decreases. Lambda was selected by assessing the lowest total root mean squared error across the dataset following iterative regression with different parameter values: 0.9 for complex spike analyses and 0.8 for simple spike analyses. Both alpha and Lambda parameters were robust across a range of values

for the distribution of coefficient weights. The same procedure was used to obtain both Purkinje cell and granule cell coefficient weights.

For analyses of both imaging and electrophysiological data, multilinear regression produced a vector of coefficient weights for all regressors for the activity of each cell/voxel. In the latter case, a separate set of coefficient weights was obtained for complex spikes and simple spikes. The estimated weights for each regressor for a given cell/voxel can take positive or negative values (or zero). Negative weights are interpretable as a relay through inhibitory neurons.

Purkinje cell maps (*Figure 2a*) shows mean z-projections of the regressor coefficients from a representative fish. Granule cell maps (*Figure 6a*) are means of seven morphed fish and were manually masked either for parallel fibers and granule cell somata to show potential differences in the signal topography. To further dissociate motor and sensory responses for sensory stimuli that strongly drive a particular behavior (translational motion and swimming, or rotational motion and left/right eye velocities), we used a maximum intensity projection of respective sensory and motor regressor maps and colored a pixel depending on whether sensory (magenta) or motor (green) regressors explain this pixel better with a given minimum distance. Differences that are below that minimum distance or are uncorrelated are colored white. Despite the slow time constant for the calcium signal decay, the variability of tail and eye movements across trials, including their onset, duration, and presence/absence, was sufficient to assign clear sensory or motor origins to the majority of these voxels.

For detailed electrophysiological analyses of the different classes of visual complex spike responses for Purkinje cells, we included for analyses all cells for which that regressor coefficient weight was significant. To determine which Purkinje cells showed significant responses to luminance, we used autocorrelation analyses of complex spike rates during whole-field flashes only and assessed significance using the Ljung-Box Q-test. For analysis of complex spikes and motor activity, we analyzed all cells with significant, nonzero motor regressor weights for complex spike activity. When examining the relationship between complex spikes and simple spike rates in individual Purkinje cells, cells with less than ten complex spikes for any condition (e.g. motor versus non-motor) were excluded from analysis.

Our multilinear regression analyses were carefully chosen in place of a series of separate simple regressions which would not provide useful or even correct insight into the question of which features these neurons are encoding. Multilinear regression is therefore preferred statistical method when considering which of multiple features contribute to a given signal and to what degree. However, as with any analysis, one must acknowledge the potential caveats or considerations when using this method (see *Slinker and Glantz, 2008* for review). For example, although multilinear regression assumes a simple addition of the regressor multiplied by the coefficient values, different sensory and/or motor features could interact nonlinearly to influence a cell's firing rate. Models do exist that incorporate nonlinearity (interaction terms), however these terms will highly correlate with each of the variables used to create the product and artificially introduce multicollinearity. Therefore since the $R^2$ values of the linear fits were reasonable, we did not explore these models. The complete set of regressors used for electrophysiological analysis nonetheless face the consideration that even in a linear model some regressors will be correlated with each other (for example, stimulus onset and duration, or swim strength and duration). We addressed this concern in two ways. First, we explored a wide range of possible regressors, both quantitative and categorical, then we dropped unnecessary and redundant regressors that consistently gave small or zero coefficient values. This was done through variable selection methods to select the optimal pool of regressors. Second, we used the elastic net optimization of lasso with low alpha values that approach ridge regression, which specifically helps to sparsify the coefficients rather than split coefficient weights between correlated regressors.

## Principal component analysis (PCA)

We performed PCA on the vector of correlations with all regressors for all automatically segmented ROIs and all fish. This correlation vector representation was clustered in the PC space in 10 clusters using *k*-means. This number was chosen because 10 PCs already explained ~90% of the variance. All voxels were then colored in according to the cluster they belonged to.

The anatomical clustering and stereotypy indices were calculated as follows. For the anatomical clustering index, the average distance between ROIs of the same cluster within a fish was compared

against the average distance between an ROI from that cluster and a randomly chosen ROI from that fish. The inverse ratio of these two quantities is the anatomical clustering index. The stereotypy index is computed similarly. In this case, the average distance between an ROI from a particular cluster and fish and other ROIs from that same cluster but other fish is compared against the distance between an ROI from that same cluster and fish and other ROIs from other clusters and fish. Again, the inverse ratio of these two quantities is the stereotypy index. To summarize, the index is a comparison of average distance within a condition to average distance without the restraint of that condition.

## Purkinje cell morphology

Sparsely labelled Purkinje cells were imaged using a 20x water immersion objective with 1 NA (Zeiss) on a confocal microscope (LSM 700, Carl Zeiss, Germany). High resolution stacks of Purkinje cells were deconvolved using Richardson-Lucy algorithm and artifacts were removed manually. Purkinje cell axonal projections were traced using NeuTube (*Feng et al., 2015*) and the Simple Neurite Tracer plugin for FIJI (*Longair et al., 2011*). SWC files were converted to line stacks and post-processed using custom written software in Python. Individual axonal projections were morphed together to a reference brain using *aldoca*:gap43-mCherry as a reference and CMTK as morphing tool (*Rohlfing and Maurer, 2003*). Dendritic planarity was assessed by performing principal component analysis on binarized dendritic morphologies. The ratio of the third principal component to the second was used to determine planarity (planar dendrites have ratios approaching 0, whereas nonplanar dendrites have ratios approaching 1).

## Purkinje cell counting

We imaged three individual PC:NLS-GCaMP6s transgenic fish line at seven dpf using confocal microscopy as described for morphological experiments above. In this line, GCaMP6s is restricted to the nucleus and approximates a sphere. Consequently, we used 3D template matching using a 3D (spherical) Gaussian to find individual nuclei using custom written software in Python. False positives were removed and missed cells were added manually.

## Quantification and statistical analysis

Data were analyzed in MATLAB and Python with custom software (*Knogler, 2019*; copy archived at https://github.com/elifesciences-publications/Knogler_etal_2019_eLife).

Values given in the text are mean ± standard error of the mean. Baseline complex spike firing rates for groups of Purkinje cells sorted by complex spike phenotype were compared by one-way ANOVA, followed by pairwise *post hoc* analyses using Bonferroni post hoc correction. The nonparametric Wilcoxon signed rank test was used on paired nonparametric datasets. Details of statistical analyses are found in the text and figure legends.

## Data/resource sharing

Example electrophysiological datasets are available at https://zenodo.org/record/1494071. An example imaging dataset is available at https://zenodo.org/record/1638807. Further information and requests for data, resources, and reagents should be directed to Ruben Portugues (rportugues@neuro.mpg.de).

## Acknowledgements

We thank Reinhard Köster for kindly providing the ca8 backbone used for transgenic line development. We thank Herwig Baier for providing mn7GFF;UAS:GFP and UAS:GCaMP6s fish. We thank Oliver Griesbeck for use of a Photometrics Prime 95B camera. We thank David Herzfeld, Vilim Štih, and Éliane Proulx for helpful comments on the manuscript. We thank the Reviewing Editor Indira Raman for useful advice. LDK was funded by the Alexander von Humboldt Foundation, the Carl von Siemens Foundation, the Fonds de recherche du Québec - Santé, and the Max Planck Gesellschaft (MPG). AMK was funded by the International Max Planck Research School for Life Sciences (IMPRS-LS), a Joachim-Herz fellowship, and the MPG. RP was funded by the MPG. This research was also partly funded by the DFG (Deutsche Forschungsgemeinschaft - German Research Foundation)

through grant PO 2105/2–1. LDK would like to dedicate this paper to the memory of her colleague and friend, Dr. Sean E Low.

## Additional information

### Funding

| Funder | Grant reference number | Author |
|---|---|---|
| Alexander von Humboldt-Stiftung | | Laura D Knogler |
| Carl Friedrich von Siemens Foundation | | Laura D Knogler |
| Fonds de Recherche du Québec - Santé | | Laura D Knogler |
| Max-Planck-Gesellschaft | Open-access funding | Laura D Knogler Andreas M Kist Ruben Portugues |
| International Max Planck Research School for Life Sciences | | Andreas M Kist |
| Joachim Herz Stiftung | | Andreas M Kist |
| Deutsche Forschungsgemeinschaft | PO 2105/2-1 | Laura D Knogler Andreas M Kist Ruben Portugues |

The funders had no role in study design, data collection and interpretation, or the decision to submit the work for publication.

### Author contributions

Laura D Knogler, Conceptualization, Software, Formal analysis, Supervision, Funding acquisition, Investigation, Methodology, Writing—original draft; Andreas M Kist, Resources, Software, Formal analysis, Funding acquisition, Investigation, Methodology, Writing—original draft, Writing—review and editing; Ruben Portugues, Conceptualization, Software, Formal analysis, Supervision, Funding acquisition, Investigation, Methodology, Writing—original draft, Project administration

### Author ORCIDs

Laura D Knogler https://orcid.org/0000-0001-7111-9832
Andreas M Kist http://orcid.org/0000-0003-3643-7776
Ruben Portugues http://orcid.org/0000-0002-1495-9314

### Ethics

Animal experimentation: All procedures involving animals were in accordance with the Max Planck Society guidelines and approved by the Regierung von Oberbayern (TVA# 55-2-1-54-2532-82-2016)

### Decision letter and Author response

Decision letter https://doi.org/10.7554/eLife.42138.024
Author response https://doi.org/10.7554/eLife.42138.025

## Additional files

### Supplementary files

• Transparent reporting form
DOI: https://doi.org/10.7554/eLife.42138.022

## Data availability

Example electrophysiological datasets are available at https://zenodo.org/record/1494071. An example imaging dataset is available at https://zenodo.org/record/1638807. MATLAB code for electrophysiological analysis available via GitHub (https://github.com/portugueslab/Knogler_etal_2019_eLife; copy archived at https://github.com/elifesciences-publications/Knogler_etal_2019_eLife).

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
