## [Decision Letter]

[Editors’ note: this article was originally rejected after discussions between the reviewers, but the authors were invited to resubmit after an appeal against the decision.]

Thank you for submitting your work entitled "Motor context dominates output from Purkinje cell functional regions during reflexive visuomotor behaviors" for consideration by *eLife*. Your article has been reviewed by three peer reviewers, and the evaluation has been overseen by a Reviewing Editor and a Senior Editor. The following individuals involved in review of your submission have agreed to reveal their identity: Eva Naumann (Reviewer #1); David L McLean (Reviewer #2).

Our decision has been reached after extensive consultation among the reviewers and editors. Based on these discussions and the individual reviews below, we regret to inform you that your work in its present form will not be considered further for publication in *eLife*.

Because the depth of the discussion is not fully represented in the reviews, we provide a fuller than usual explanation for the decision, quoting and paraphrasing from the discussion (including editor's comments), in the hope that it will be of value to you as you proceed with your work. As you will see in the reviews, the reviewers found a great deal of value in the work from the perspective of the technical achievement of documenting the climbing fiber responses of Purkinje cells of the full zebrafish cerebellum in the context of visuomotor behaviors. There were, however, some important concerns, as follows:

1) Relating to approach and analysis: there is much potentially valuable data, but analyses are opaque and not clearly validated. Most simply, we are not actually shown the relationships between the data and the transformations sufficiently to be assured of their validity. We quickly are taken to derived measures, but methods are not always clear, critical tests of the results are absent, considerations of context and alternatives are not given, caveats are not presented, and many key elements of the data are too microscopic or on too slow of a time base to be intelligible. A few examples: a 1600 ms GCaMP convolution kernel for the imaging is long; how do we know that the correlations are valid without seeing what happens in this transformation? The windows of 'sensory' and 'motor' for the relation of complex spikes to one or the other category aren't clearly defined; since complex spikes often occur singly, it is hard to associate them with a sensory or motor event without some temporal criterion; and the swimming bouts are impossible to see on the time scales shown, and so how was a criterion defined to be associated with ‘motor’? Similar points arise for most figures. Likewise, panels are not always called out correctly, and descriptions in the text are often difficult to see reflected in the data that are illustrated.

2) Because the experiments only entail correlations within one modality, the conclusions are limited to providing additional support for attributes of cerebellar physiology for which there is already substantial accumulating evidence (e.g., complex spikes can relay sensory inputs, while granule motor-related information can come in through the granule cells; the notion of state-dependence in Figure 5 is interesting, but the presented result can simply be inferred from the idea that there is more parallel fiber drive during swimming, and so the illustration does not actually provide any further insight). In this context, attention was drawn by the reviewers to the fact that optogenetic tools that might be useful for manipulations are described in the Materials and methods but are not actually used in the paper.

3) Given that a single modality was studied, the conclusions are rather over-generalized without consideration of what the cerebellum might do in other behaviors and the final model offers little further insight., e.g., the claim in the discussion that complex spikes don't encode predictive motor signals. Nothing studied here actually tests anything about prediction. Here a relevant point might be the link to the associative learning study of Harmon et al., which found different patterns of complex and simple spike activity in a different set of behaviors, yet evidence for a topographic segregation into three classes is nevertheless still present. More experiments and/or more consideration of published work would be necessary to make the model sophisticated and flexible enough to be useful.

A strong case was made by some reviewers that perhaps more data can be gathered to conduct more investigative tests and generate some more definitive conclusions and/or a more in-depth model, which could strengthen the manuscript. If this is the case, we encourage you to address all the comments herein and appeal to resubmit the manuscript; however, the charge of the evaluation is to judge the data presented and so we cannot return a favorable decision based on the conjecture that such experiments might be forthcoming.

*Reviewer #1:*

This manuscript presents new insights on how the entire vertebrate cerebellum activity is related to multiple visuomotor transformations, particularly its relation to visual motion cues and eye and body movements. The cerebellum is thought to integrate sensory and motor-related information in Purkinje cells (PC), the main computational components that disseminate information to the output nuclei and enable motor learning.

The significance of Knogler et al.'s research lies in examining the complex relationship between sensorimotor states and PC activity (both simple spikes (SSs) and complex spikes (CSs)). They did this using an interesting multilinear regression approach that allowed them to disentangle the encoding of sensory and motor variables, even when correlated. They harness the experimental power of the zebrafish to comprehensively describe all PCs within individuals to reveal the functional topography, and intricately demonstrate how the zebrafish cerebellum segregates into functional modules, such that PCs in different regions collect visual input via climbing fibers; they also present evidence that CSs can change SS rates in a behavior-state dependent fashion.

Starting with two-photon calcium imaging, they use transgenic fish expressing fluorescent calcium sensors only in PCs to holistically describe PC function across the entire cerebellum. They reveal the response profiles of the entire set of these particular cell types. They demonstrate that anatomical functional clustering exists along behavioral dimensions, sorting PCs into three regions that handle different classes of sensory information. In a second phase of experiments, they use electrophysiology to better parse the CS and SS correlations with sensorimotor variables, and one another. The context-dependence of the CS/SS interactions was a very interesting and provocative result, with the potential to open up whole new lines of research. The inclusion of eye-movement signals as a regressor was an important control. Together, the methodologies and tools that were developed in this study are a significant step to investigate how PC function in these different zones affects learning.

The research appears rigorous, well executed and presented data are extensive and appropriately analyzed with relevant statistical methods, including appropriate treatment of individual animals. The figures are relatively easy to absorb, and while the legends are clear they could be shortened significantly. The presented supplementary data answers for many concerns and provides controls. And the Materials and methods section is clear but seems to have an unnecessary section discussing optogenetics. It seems like the authors generated (PC:ChR2(H134R)-tagRFP) under ca8 promoter control but we are not shown any data.

Despite the many strengths of the current manuscript, some questions remain and require revisions or discussion in the paper:

1) The main experimental weakness is that they focused on correlations only, without doing experimental perturbations for causality. For future directions the authors might use laser ablations or optogenetic perturbations of these different sectors or functional cell types. Evidently, one strength of their zebrafish preparation is that they can actually control (almost) all the cells in a population of interest and look at causality by measuring motor output. Since the authors seemed to have generated the optogenetic tools for this project ca8 (PC:ChR2(H134R)-tagRFP), why was no data presented? Please explain why this section was included but none of the results.

2) The authors could have done more to explain the 'voxel/multilinear regression' analysis method and it is confusing to call the imaging volumetric without explanation in the main text. In particular, it would help if they discussed the multilinear regression method in more detail: for instance, when sensory and motor variables are highly correlated (which they clearly are), how does the multilinear regression serve to separate out these effects, and does it take into account interaction terms? Also, it would be nice if they did some kind of internal validation of this technique, for instance fixing the stimulus (e.g., leftward motion) and examining how well the model predicted behavior, or at least explained why such a validation procedure was not used.

3) It would be great if they had expanded on the broader view in the literature that CSs encode error. As is, the reader learns almost nothing about how they interpret these results.

4) Another limitation is that the authors focused only on one modality without discussing relevance to other modalities and motor output. It would be better if this were discussed in more detail in both the main text and the discussion. And, in the future, it would be great to see how the same set of PCs processes information from other modalities to guide behavior.

5) Given the presented data, the model in Figure 6 is somewhat disappointing. A cell-type specific model that would illustrate the predictions for perturbation experiments would be preferred.

6) Since much of the authors' interpretation is concerned with a difference in motor and sensory signals, a different analysis than these simplified motor regressor would be favored. It would be good to show continuous quantitative readout of behavior as a regressor. Perhaps some measure like the rectified ratio of left and right outputs of the ventral root traces (as in Dunn et al., 2016). Their relatively simple measures of behavior could bias the results to favor a sensory coding scheme.

7) It would be interesting to see a map as in Figure 2F for all recorded PC neurons by using the information from electrophysiologically recorded neurons.

If these points would be addressed, the presented research is a good fit for *eLife*. Both the insights into vertebrate cerebellar function and technical execution are of high quality, justifying publication in this journal.

*Reviewer #2:*

The work by Knogler and colleagues uses sophisticated microscopic and analytical methods to examine the functional organization of Purkinje cells in the larval zebrafish cerebellum. Contrary to the authors assertions (Results section first paragraph), this is not the first population level study of the cerebellum (Aizenberg and Schuman, 2011; Matsui et al., 2014). Also, the discovery of caudolateral Purkinje cells that are associated with eye movements and more rostromedial neurons associated with tail movements has been reported previously (Matsui et al., 2014). Regression and principal component analysis has been used to dissect relative sensory and motor contributions to responses, however the there is insufficient detail describing the methodology in the main text or Materials and methods section for non-computational neuroscientists to decipher what is going on. In Figure 1C, for example, the 'for motion' regressors have three peaks that align well with three of the largest z-score deviations, but many of the other deviations appear to be ignored. Why? Unfortunately, this acts to undermine much of the paper, which is further exacerbated by occasions where the authors either call out figure panels with statements that lack supporting evidence (see Results section, third paragraph and Figure 1D – reverse grating motion and luminance responses look identical to me; subsection “Purkinje cells in different regions receive climbing fiber input related to different, specific visual features and send outputs to different downstream regions” and Figure 2D,G – I'm not sure where to find the analysis of depression here; subsection “Motor activity is broadly represented in granule cell signals” and Figure 4I – not seeing any periodicity to speak of) or are too small to make any meaningful interpretation (see the same section and Figure 4K – not sure how you are supposed to see phase differences at this magnification). From reading the manuscript, you would think that the zebrafish cerebellum is only involved in encoding visuomotor behaviors (subsections “Purkinje cells combine sensory and motor information from distinct inputs” and “Regional functionality of the zebrafish and mammalian cerebellum”). Auditory, tactile, thermosensory modalities are also highly salient sensory modalities, but are not explored here. Since these modalities were not tested, it is not clear how the pattern generalizes. The authors admit themselves that patterns can change based on how the fish is stimulated (subsection “Motor activity is broadly represented in granule cell signals”). It is also not clear how these data fit with recent descriptions of regionalization also based on complex spike activity in zebrafish (Harmon et al., 2017). Overall, I wanted to like this paper, but the lack of work made for a rather frustrating read. The authors need to do a better job of placing their work in the context of previous discoveries, focus more on what is new here, and do a more thorough job of describing it.

*Reviewer #3:*

Reading the manuscript "Motor context dominates output from Purkinje cell functional regions during reflexive visuomotor behaviors", Knogler et al. present large datasets from whole population Purkinje cell Ca^2+^-imaging and single cell patch recording in the zebrafish larval cerebellum during visually driven reflexive behaviors. While complex spikes are identified to be associated with sensory correlates, simple spike firing instead represents motor variables. Complex spikes seem to represent sensory climbing fiber input relating to visual cues and these complex spikes can modulate simple spike firing extensively, while the zebrafish larvae rest, whereas during swimming, during which simple spike firing increases gradually in frequency, modulation of simple spike firing by complex spikes is temporally more restricted to about 50 milliseconds across the Purkinje cell population. Gradually increasing simple spike firing correlates with increased swim strength but is not in phase with the latter, arguing for simple spike firing to represent motor afferent copies transmitted via granule cells rather than proprioceptive or lateral line derived input. In addition, the analysis of complex spike firing simultaneous to monitoring behavior reveals that complex spike firing precedes swim behavior but does not seem to predict motor behavior as spike firing remains unchanged albeit the stimuli-elicited swim bouts vary. Moreover, topographical analysis of complex and simple spike firing reveals a topographical organization of the Purkinje cell layer being subdivided into three functional regions along its rostro-caudal axis with respect to sensory climbing fiber input-stimulated complex firing in Purkinje cells. In contrast, motor-related simple spike firing occurs across the Purkinje cell layer without obvious topographical organization in functional regions.

Thus, in summary, this manuscript presents a huge amount of novel highly interesting and exciting findings, the analysis is performed with large care, the data is presented beautifully and manuscript clearly and convincingly written. Not being an expert in mathematics, computation and electrophysiology I cannot comment extensively on this part of the manuscript and have to leave this to my colleague reviewers. From the point of neuroanatomical organization of the differentiating zebrafish cerebellum some points remain that could in my opinion make this manuscript even nicer:

1) The authors suggest that the Purkinje cell population receives a topographically organized input from climbing fibers stimulating complex firing localized to three different regions along the rostro-caudal axis of the Purkinje cell layer (e.g. in paragraph five of subsection “Purkinje cells in different regions receive climbing fiber input related to different, specific visual features and send outputs to different downstream regions”). Climbing fiber projections in zebrafish have been demonstrated but are only poorly characterized at last. I understand that the authors are working on this and a full neuroanatomical and physiological analysis of climbing fiber projection and connectivity will exceed the purpose and focus of this manuscript. Nevertheless, a few studies supporting the suggested topographic climbing fiber organization should be provided.

2) In addition, it remained unclear to me whether climbing fibers synapse with Purkinje cell somata or along their dendritic trees or both. Is there a preferential location of PC-CF synapses?

3) The authors state the zebrafish contain two different types of anatomically different Purkinje cells according to their projections that occur either internally to eurydendroid cells or externally to the vestibular nuclei. Does this model also include Purkinje cell collaterals projecting to Purkinje cells nearby or do the authors think that such PC-PC connectivities do not exist in zebrafish?

4) The authors performed whole cell recordings in the caudolateral region of the Purkinje cell layer during rotational windmill motion evoked optokinetic reflex. Were these Purkinje cells internally of externally projecting PCs?

---

## [Author Response]

Because the depth of the discussion is not fully represented in the reviews, we provide a fuller than usual explanation for the decision, quoting and paraphrasing from the discussion (including editor's comments), in the hope that it will be of value to you as you proceed with your work. As you will see in the reviews, the reviewers found a great deal of value in the work from the perspective of the technical achievement of documenting the climbing fiber responses of Purkinje cells of the full zebrafish cerebellum in the context of visuomotor behaviors. There were, however, some important concerns, as follows:1) Relating to approach and analysis: there is much potentially valuable data, but analyses are opaque and not clearly validated. Most simply, we are not actually shown the relationships between the data and the transformations sufficiently to be assured of their validity. We quickly are taken to derived measures, but methods are not always clear, critical tests of the results are absent, considerations of context and alternatives are not given, caveats are not presented, and many key elements of the data are too microscopic or on too slow of a time base to be intelligible. A few examples: a 1600 ms GCaMP convolution kernel for the imaging is long; how do we know that the correlations are valid without seeing what happens in this transformation? The windows of 'sensory' and 'motor' for the relation of complex spikes to one or the other category aren't clearly defined; since complex spikes often occur singly, it is hard to associate them with a sensory or motor event without some temporal criterion; and the swimming bouts are impossible to see on the time scales shown, and so how was a criterion defined to be associated with ‘motor’? Similar points arise for most figures. Likewise, panels are not always called out correctly, and descriptions in the text are often difficult to see reflected in the data that are illustrated.

We apologize that the regression analysis was not explained in sufficiently comprehensive detail. We wrongfully assumed that people would be familiar with this approach and we regret that this made the manuscript so difficult to follow.

We have significantly expanded the details in the Materials and methods section to better explain the analysis and parameter validation. The main text also offers a better introduction to the analysis and context for the choice of analysis. References have been added for these analyses as applied to other relevant datasets. We specifically mention caveats and other considerations for these analyses as a paragraph in the Materials and methods, including the discussion of interaction terms, variable selection methods, and elastic net optimization.

This expansion of the analysis explanation in the main text furthermore includes a revision of the figures such that Figure 1 is now an overview of the experiment and regression methodology. In particular, Figure 1D now shows an overview of the regression process, while Figure 1E shows an example calcium signal that is recapitulated by the summation of several regressors scaled by their coefficient weights. This latter panel also helps show how sensory and motor signals can be clearly separated.

We now provide the full set of regressors for both imaging (Figure 1—figure supplement 1) and electrophysiology (Figure 3—figure supplement 1) as these are slightly different (as explained in the Materials and methods). A new figure supplement (Figure 3—figure supplement 2) also further shows how visual and motor activity can be resolved with electrophysiological signals.

In the Materials and methods, we better explain our reasons for choosing the regressors and time windows. We are confident that the temporal time windows are reasonably chosen despite the relatively low complex spike rate because we clearly find complex responses associated with visual features including motion onset and luminance, for which a 500 ms bin at stimulus onset captures these responses very well. Swim bouts are on the order of 500 ms in duration, therefore this should be sufficiently long to capture a bout-related complex spike response, yet these are very rare despite looking for a relationship to the graded swim strength, swim duration, or the phase of spiking activity in relation to swim frequencies. We also look for bout-related responses preceding or following a bout, with another window, thereby providing comprehensive coverage around a bout for a motor-related complex spike to be found.

Finally, we have been careful to revise the figures and text throughout to ensure that all panels are sufficiently large to be readable, and that the text calls these panels correctly and in a way that is easy to follow.

2) Because the experiments only entail correlations within one modality, the conclusions are limited to providing additional support for attributes of cerebellar physiology for which there is already substantial accumulating evidence (e.g., complex spikes can relay sensory inputs, while granule motor-related information can come in through the granule cells; the notion of state-dependence in Figure 5 is interesting, but the presented result can simply be inferred from the idea that there is more parallel fiber drive during swimming, and so the illustration does not actually provide any further insight). In this context, attention was drawn by the reviewers to the fact that optogenetic tools that might be useful for manipulations are described in the Materials and methods but are not actually used in the paper.

Although our findings only pertain to the visual modality, we provide a comprehensive mapping of this modality across the entire Purkinje cell population. This has not been achieved in mammalian cerebellum for any modality. Furthermore, because of the nature of the visual stimuli we used and the variable eye and tail behaviors that are elicited, there are many components to the visual stimuli and motor responses. As a result, we are not limited to a simple first-order classification of complex spikes as sensory or motor (which many people have done previously in various contexts) but instead we can say which particular aspect of the sensory stimulus or motor response they relate to. For instance, we see graded complex spike responses to rotational velocity for some cells, and a transient response to stimulus onset for others. Our findings therefore show important qualitative and quantitative differences in the complex coding of visual features across regions of Purkinje cells that can now be used as a foundation for future manipulations (e.g. optogenetic experiments).

With respect to optogenetics in particular, it was our mistake to leave the mention of optogenetic tools in the Materials and methods. We have indeed developed some genetic tools for Purkinje cell optogenetics. However, the effect of channelrhodopsin in Purkinje cells is complex and many more controls are needed to understand and/or optimize these experiments. For example, due to the bistability of Purkinje cells (sensitivity to ionic balance), channelrhodopsin can easily lead to depolarization block of simple spikes in Purkinje cells rather than driving simple spikes, and the levels of power need to be carefully titrated for individual cells. We and no doubt others will follow up this work with experiments to probe manipulations of activity in these regions and with respect to behavior, however this work is beyond the scope of the current paper. We therefore decided, in consultation with the editors, not to add data but to instead clarify what we already have, as these findings already represent a large amount of data and analyses for the reader to take away from the current manuscript.

With regards to the comment that “the notion of state-dependence in Figure 5 is interesting, but the presented result can simply be inferred from the idea that there is more parallel fiber drive during swimming,” we would like to point out that these findings are not necessarily what is expected from the literature. Sengupta and Thirumalai, 2015, working in larval zebrafish, suggested that a spontaneous complex spike “toggles” the simple spike rate of a Purkinje cell, such that an “up state” (i.e. high simple spike rate) is ended by a complex spike and sent to the “down state,” and vice versa. We in fact do not see this effect during visuomotor behaviors, suggesting that the interaction of a complex spike with simple in a motor context has different network/plasticity effects as we discuss in the text.

Finally, in further support of the notion that the interplay between complex spikes and behavioral context is an important and relevant feature for understanding cerebellar function, we highlight in the discussion recent work from the Carey lab in mouse cerebellum showing that reflexive locomotor activity or a broad but mild optogenetic activation of granule cells enhances cerebellar learning.

3) Given that a single modality was studied, the conclusions are rather over-generalized without consideration of what the cerebellum might do in other behaviors and the final model offers little further insight., e.g., the claim in the discussion that complex spikes don't encode predictive motor signals. Nothing studied here actually tests anything about prediction. Here a relevant point might be the link to the associative learning study of Harmon et al., which found different patterns of complex and simple spike activity in a different set of behaviors, yet evidence for a topographic segregation into three classes is nevertheless still present. More experiments and/or more consideration of published work would be necessary to make the model sophisticated and flexible enough to be useful.

We regret that we over-generalized the conclusions from this singular modality. We have altered the text in several ways to better address these concerns.

Firstly, we now discuss other modalities including auditory, vestibular, olfactory and lateral line. In particular, we discuss evidence for cerebellar coding of these modalities from calcium imaging studies and we cite data showing that cerebellar-dependent associative learning with the auditory modality in unsuccessful at this stage (and even absent in the 6 week-old larva). This in particular suggests that the visual modality is more developed than audition at this stage. We also cite very recent findings from two studies published this month that show cerebellar activity strongly modulated by vestibular inputs, and relate this together with our findings for the visual coding of rotational motion to the known mammalian cerebellar-dependent vestibulo-ocular reflex.

Next, the findings of the Harmon et al. study using a different sensorimotor paradigm are now better utilized and integrated into our results and discussed in multiple contexts. Briefly, we make direct links to the similar regionality of Purkinje cell complex spike phenotypes in their associative learning paradigm. We also comment on the finding from their study that the central region of the cerebellar cortex appears to acquire novel complex spike signals during conditioning that complements the distinctively more flexible encoding of innate features in this region in our current study.

We have confidence that the visual organization of the cerebellum we observe is important. Although only the visual modality was tested, the degree to which nearly all Purkinje cells fit into these three observed complex spike visual phenotypes is striking. Nevertheless, in addition to the considerations above, we explicitly state that other modalities certainly need to be tested, to determine if/how they map onto these regions, and that the observed mapping may also change with development.

Additionally, we have rephrased the text discussing our interpretation of the meaning of these complex spikes. When we originally stated they were not predictive, we meant this in the sense that a complex spike occurring for a visual stimulus does not predict an upcoming episode of behavior in response to that stimulus. We have now clarified this in an expanded discussion that furthermore talks about the “error” and “novelty” hypotheses for complex spikes.

Reviewer #1:[…]Despite the many strengths of the current manuscript, some questions remain and require revisions or discussion in the paper:1) The main experimental weakness is that they focused on correlations only, without doing experimental perturbations for causality. For future directions the authors might use laser ablations or optogenetic perturbations of these different sectors or functional cell types. Evidently, one strength of their zebrafish preparation is that they can actually control (almost) all the cells in a population of interest and look at causality by measuring motor output. Since the authors seemed to have generated the optogenetic tools for this project ca8 (PC:ChR2(H134R)-tagRFP), why was no data presented? Please explain why this section was included but none of the results.

We apologize for having mistakenly included this transgenic line in the Materials and methods section. As discussed above, this tool is not yet sufficiently optimized for manipulation experiments. Purkinje cells are a rather special type of neuron and although in the future we are planning experimental manipulations including ablations and optogenetic perturbations to probe the role of simple spikes and complex spikes in these behaviors, there is a considerable amount of groundwork to do before this is possible. The editors support our decision to instead focus on the current dataset. Although the data does not directly probe causality, it nonetheless provides a wealth of new knowledge and hypotheses for future work in the zebrafish cerebellum as well as insight into the general principles of feature encoding and organization that should extend to other species.

2) The authors could have done more to explain the 'voxel/multilinear regression' analysis method and it is confusing to call the imaging volumetric without explanation in the main text. In particular, it would help if they discussed the multilinear regression method in more detail: for instance, when sensory and motor variables are highly correlated (which they clearly are), how does the multilinear regression serve to separate out these effects, and does it take into account interaction terms? Also, it would be nice if they did some kind of internal validation of this technique, for instance fixing the stimulus (e.g., leftward motion) and examining how well the model predicted behavior, or at least explained why such a validation procedure was not used.

The term volumetric in the context of two-photon imaging is often used to refer to two-photon imaging at successive planes in a sample to yield a volume (Renninger and Orger, 2013; Portugues et al., 2014). In the Materials and methods under ‘Functional population imaging’ we nonetheless made this explicit by saying “We acquired the total cerebellar volume by sampling each plane at ~ 5 Hz. After all stimuli were shown in one plane, the focal plane was shifted ventrally by 1 μm and the process was repeated.” We now also mention these details in the main Results text for clarity. We moreover would like to note that this method (compared to e.g. confocal or light sheet microscopy) allows for very high, single-cell resolution. The additional of Figure 1—figure supplement 1 panels, showing an anatomical stack along with planewise regressor maps, and Video 2 showing the activity from a single two-photon plane, should also clarify this for readers.

The lack of details given the multilinear regression were a major issue for all reviewers. Please see the earlier Reviewing Editor comments, point 1, for our response to this issue.

Please also note that visual and motor features are not highly correlated – see additional text, Materials and methods, and figures including Figure 1E and Figure 3—figure supplement 2.

We do not include interaction terms in the multilinear regression model as the model without them had good fits and introducing these complicated additional terms can produce misleading results that would be very difficult to interpret. This is clarified in the Materials and methods.

In response to your last comment regarding an internal validation, we would like to say that we are not trying to predict behavior from any model and are simply using the regression to describe the encoding of stimulus and motor variables in the Purkinje cell activity. We hope that the expanded section on regression clarifies this.

3) It would be great if they had expanded on the broader view in the literature that CSs encode error. As is, the reader learns almost nothing about how they interpret these results.

The Discussion has been edited considerably and now contains a section titled ‘What signals are complex spikes encoding?’ in order to better explain our interpretation of the complex spike results in relation to several current hypotheses including the “error” hypothesis. We would furthermore like to mention that one of the strengths of our study is that we see consistent signals across experimental paradigms including when the fish is fully paralyzed, trunk-embedded with eyes freed, and fully eye and tail-freed. Thus, we are able for example to directly evaluate and reject the interpretation of complex spikes during rotational motion as an “error” signal caused by retinal slip.

4) Another limitation is that the authors focused only on one modality without discussing relevance to other modalities and motor output. It would be better if this were discussed in more detail in both the main text and the discussion. And, in the future. it would be great to see how the same set of PCs processes information from other modalities to guide behavior.

The discussion of other modalities was an important point raised by many reviewers and we agree that we are interested in exploring this in the future. Please see the earlier Reviewing Editor comments, point 3, for our response to this issue.

5) Given the presented data, the model in Figure 6 is somewhat disappointing. A cell-type specific model that would illustrate the predictions for perturbation experiments would be preferred.

We provided the schematic in this final figure as a summary figure rather than a predictive model. Since the data presented in this manuscript is dense, we thought that such a summary would be a helpful graphic overview of our findings. Without any results from perturbation experiments, a predictive model would be highly speculative.

6) Since much of the authors' interpretation is concerned with a difference in motor and sensory signals, a different analysis than these simplified motor regressor would be favored. It would be good to show continuous quantitative readout of behavior as a regressor. Perhaps some measure like the rectified ratio of left and right outputs of the ventral root traces (as in Dunn et al., 2016). Their relatively simple measures of behavior could bias the results to favor a sensory coding scheme.

A continuous quantitative readout of behavior (“vigor”) was already a motor regressor (as shown in updated Figure 1—figure supplement 1, Figure 3—figure supplement 1 and 2 and mentioned in the text and Materials and methods). We have greatly expanded the explanation of both the sensory and motor regressors and the analysis in the text, Materials and methods, and figures to ensure that this is clearer for the readers.

During functional imaging experiments, the tail movement is used to extract these motor features. For electrophysiology, it is the ventral root recording. We cannot provide the rectified ratio of left and right ventral root traces as suggested because we perform unilateral recordings from the left ventral root only. However, we find no rhythmicity in the Purkinje cell complex spikes or simple spikes that suggest a patterning of input in line with swimming frequencies (autocorrelation analyses, updated Figure 5F). We do however compute an integrated quantitative readout of swim strength from the single ventral root recording in order to obtain the readout of bilateral swim vigor. We furthermore are satisfied that this captures well the behavioral dynamics since the simple spike rates are highly correlated with this continuous vigor regressor.

7) It would be interesting to see a map as in Figure 2F for all recorded PC neurons by using the information from electrophysiologically recorded neurons.

The map in Figure 3F (previously Figure 2F) does indeed show the location of the 61 Purkinje cells recorded by electrophysiology.

Reviewer #2:The work by Knogler and colleagues uses sophisticated microscopic and analytical methods to examine the functional organization of Purkinje cells in the larval zebrafish cerebellum. Contrary to the authors assertions (Results section first paragraph), this is not the first population level study of the cerebellum (Aizenberg and Schuman, 2011; Matsui et al., 2014). Also, the discovery of caudolateral Purkinje cells that are associated with eye movements and more rostromedial neurons associated with tail movements has been reported previously (Matsui et al., 2014).

We have toned down this assertion however we wish to emphasize that our work is not replicating the previous studies you cite.

1) The first reference, Aizenberg and Schuman, use in their study a bolus injection of organic calcium indicators to image local groups of mixed neuronal cell types including eurydendroid cells, Purkinje cells, and molecular layer interneurons. They themselves state that this is a mixed population. The use of the injection method and the fact that they furthermore only imaged five focal planes means that these imaging results are neither comprehensive nor exclusive across the Purkinje cell population. We nonetheless appreciate the insight from this work and cite this work in the current manuscript.

2) The imaging for the Matsui study was done on a confocal microscope with the pinhole maximally open, therefore the resolution was very low. Signals are also much weaker in the GCaMP5g transgenic line than in the GCaMP6s line both in general and specifically in Purkinje cells (personal observations and communications). Our imaging results therefore provide a large improvement in resolution that significantly expand and refine these regions. We also have an expanded set of visual stimuli that is significant for the novel insights we present in our current study regarding the overarching visual organization of the cerebellum.

Furthermore, without electrophysiological data to support the calcium imaging, no conclusions could be made from Matsui et al. regarding the origin of the calcium signals from simple spikes or complex spikes. We believe their findings represent a mix of sensory complex spike and motor simple spike contributions that we only now disambiguate with our new results.

Our findings in particular show not only new but also substantially different results from the study indicated:

1) We report that caudolateral Purkinje cell complex spikes are *not* associated with eye movements but instead sensory (visual) motion of the grating stimulus

2) We report that the simple spike activity of nearly *all* Purkinje cells are associated with tail movements while almost no complex spikes report motor information. It is telling that nowhere in the Matsui et al. paper is the term “complex spikes” or “simple spike” mentioned, therefore mammalian cerebellar researchers would likely incorrectly attribute these signals to complex spikes.

3) We report that rostromedial Purkinje cell complex spikes are associated with directional onset of sensory motion and that central Purkinje cells heterogeneously encode luminance.

Our ability to not only properly disambiguate between not only visual and motor responses, but also attribute a complex spike or simple spike origin to individual Purkinje cell signals, is of major importance for the interpretation of cerebellar function.

Regression and principal component analysis has been used to dissect relative sensory and motor contributions to responses, however the there is insufficient detail describing the methodology in the main text or Materials and methods section for non-computational neuroscientists to decipher what is going on. In Figure 1C, for example, the 'for motion' regressors have three peaks that align well with three of the largest z-score deviations, but many of the other deviations appear to be ignored. Why?

We apologize that insufficient details were provided initially to allow for a clear reading and interpretation of the results. Please see the earlier Reviewing Editor comments, point 1, for our response to this issue.

On a more specific note, the example mentioned for what was previously Figure 1C appears to be a misunderstanding. As we stated in the main text accompanying the reference to this figure panel, “Each different feature of the sensory stimulus or motor behavior, such as translational motion in a certain direction, or the duration of the swimming bouts across a trial, were used to build a vector of values for the trial duration.”. It is for precisely the reason that more than one regressor may contribute to the calcium activity that we use the complete set of all single sensory and motor regressors in the multilinear regression. The other deviations in these example calcium traces would therefore have nonzero coefficients for other regressors beyond the one that is shown. This is now made more explicit in revised Figure 1D,E and the text and Materials and methods.

Unfortunately, this acts to undermine much of the paper, which is further exacerbated by occasions where the authors either call out figure panels with statements that lack supporting evidence (see Results section, third paragraph and Figure 1D – reverse grating motion and luminance responses look identical to me; subsection “Purkinje cells in different regions receive climbing fiber input related to different, specific visual features and send outputs to different downstream regions” and Figure 2D,G – I'm not sure where to find the analysis of depression here; subsection “Motor activity is broadly represented in granule cell signals” and Figure 4I – not seeing any periodicity to speak of) or are too small to make any meaningful interpretation (see the same section and Figure 4K – not sure how you are supposed to see phase differences at this magnification).

The figure panels are now called more carefully. For example, the depression was not obvious from the example shown in Figure 4I as you mentioned, therefore it is now quantified in the results and does not refer to a figure panel. Our apologies that some panels were indeed too small; figures have been split up and/or adjusted to allow for all panels and text to be of adequate size.

From reading the manuscript, you would think that the zebrafish cerebellum is only involved in encoding visuomotor behaviors (subsections “Purkinje cells combine sensory and motor information from distinct inputs” and “Regional functionality of the zebrafish and mammalian cerebellum”). Auditory, tactile, thermosensory modalities are also highly salient sensory modalities, but are not explored here. Since these modalities were not tested, it is not clear how the pattern generalizes.

Please see the Reviewing Editor comments, point 3, for our response to this issue.

Briefly, we discuss these points in regards to other modalities and behavioral paradigms (including that of Harmon et al.) to better place these findings in context.

The authors admit themselves that patterns can change based on how the fish is stimulated (subsection “Motor activity is broadly represented in granule cell signals”).

We do not claim that these patterns change based on how the fish is stimulated – we do not see that the responses change. What this statement meant is that our 2017 study elicited swimming behavior so rarely (<10% probability of swimming during forward gratings, due to very low contrast settings, see Figure 4A in Knogler et al., 2017) that these current responses are not directly comparable because the previous study essentially represented a no-swimming paradigm (and shock-related “escapes” are very different).

It is also not clear how these data fit with recent descriptions of regionalization also based on complex spike activity in zebrafish (Harmon et al., 2017).

Please see the Reviewing Editor comments, point 3, for comments related to this issue.

We would also like to mention to this reviewer that in the first submission we did in fact cite this work several times and drew positive parallels between it and the current study. We are however using different paradigms to answer different questions – we are trying to understand the innate feature coding for visual stimuli and motor activity while Harmon et al. focused on conditioned responses that emerged over time.

We did however fail to adequately make use of the similarity in findings regarding the regionalization of Purkinje cell responses (albeit it to different features) in the original submission and in the revised manuscript we better highlight this work and the parallels that arise with respect to complex spike regionalization.

Overall, I wanted to like this paper, but the lack of work made for a rather frustrating read. The authors need to do a better job of placing their work in the context of previous discoveries, focus more on what is new here, and do a more thorough job of describing it.

We have significantly edited the text to address these concerns. We hope that this reviewer will find the new version much clearer and more insightful.

Reviewer #3:[…]1) The authors suggest that the Purkinje cell population receives a topographically organized input from climbing fibers stimulating complex firing localized to three different regions along the rostro-caudal axis of the Purkinje cell layer (e.g. in paragraph five of subsection “Purkinje cells in different regions receive climbing fiber input related to different, specific visual features and send outputs to different downstream regions”). Climbing fiber projections in zebrafish have been demonstrated but are only poorly characterized at last. I understand that the authors are working on this and a full neuroanatomical and physiological analysis of climbing fiber projection and connectivity will exceed the purpose and focus of this manuscript. Nevertheless, a few studies supporting the suggested topographic climbing fiber organization should be provided.

The work of Bae et al. (2009) is cited in regard to the conservation of climbing fiber properties in the Introduction. We have now explicitly stated that (like in mammals) all climbing fibers cross the midline after leaving the inferior olive and thus contact Purkinje cells in the contralateral hemisphere of the cerebellum (Takeuchi et al., 2015). We furthermore added the interesting finding that although only one bundle of ipsilateral climbing fibers in seen at larval stages, in the adult other fiber bundles are visible (Takeuchi et al., 2015), suggesting that other routes or types of information are added for communication between the inferior olive and cerebellar cortex at later stages. Finally, we cite the review of Apps and Hawkes (2009) for information about the hardwired topography of climbing fiber projections to Purkinje cells in the mammalian cerebellum.

2) In addition, it remained unclear to me whether climbing fibers synapse with Purkinje cell somata or along their dendritic trees or both. Is there a preferential location of PC-CF synapses?

Takeuchi et al. (2015) find in their study that climbing fibers project onto the somata or proximal dendrites of Purkinje cells in larvae and adult animals. This has now been mentioned in the text.

3) The authors state the zebrafish contain two different types of anatomically different Purkinje cells according to their projections that occur either internally to eurydendroid cells or externally to the vestibular nuclei. Does this model also include Purkinje cell collaterals projecting to Purkinje cells nearby or do the authors think that such PC-PC connectivities do not exist in zebrafish?

The experimental techniques in our current study did not provide insight into this particular neuroanatomical question. The finding that Purkinje cells that are located very closely to each other have similar complex spike and simple spike responses, however, would suggest that even if these PC-PC collaterals exist, they are not sufficient to provide mutual spike inhibition. We will not however discuss this in the text as this interpretation is speculative.

4) The authors performed whole cell recordings in the caudolateral region of the Purkinje cell layer during rotational windmill motion evoked optokinetic reflex. Were these Purkinje cells internally of externally projecting PCs?

The partial answer to this question can be found in Figure 4E, in the revised manuscript (previously Figure 3E) and in the results. Of 17 Purkinje cells for which we had complete morphology as well as single-cell electrophysiological recordings, 6/7 Purkinje cells with externally-projecting axons (referred to as “caudal axonal projection” in the figure panel) showed these responses to rotational windmill motion (and only 1/10 Purkinje cells with internal axons had this phenotype). In addition, there is indirect evidence to support this at the population level, based on the spatial overlap of Purkinje cells with externally-projecting axons and the complex spike phenotype for rotational windmill motion (compare Figure 2B, Figure 3F, Figure 4F, Figure 4—figure supplement 1H).